# Mechanism of life-long maintenance of neuron identity despite molecular fluctuations

**Joleen JH Traets[1†], Servaas N van der Burght[2‡], Suzanne Rademakers[2], Gert Jansen[2], Jeroen S van Zon[1]\***

[1]Department of Living Matter, AMOLF, Amsterdam, Netherlands; [2]Department of Cell Biology, Erasmus University Medical Centre, Rotterdam, Netherlands

**\*For correspondence:**
j.v.zon@amolf.nl

**Present address:** [†]Division of Tumor Biology & Immunology and Division of Molecular Oncology and Immunology Oncode Institute, Netherlands Cancer Institute, Amsterdam, Netherlands; [‡]The Gurdon Institute, University of Cambridge, Cambridge, United Kingdom

**Competing interest:** The authors declare that no competing interests exist.

**Abstract** Cell fate is maintained over long timescales, yet molecular fluctuations can lead to spontaneous loss of this differentiated state. Our simulations identified a possible mechanism that explains life-long maintenance of ASE neuron fate in *Caenorhabditis elegans* by the terminal selector transcription factor CHE-1. Here, fluctuations in CHE-1 level are buffered by the reservoir of CHE-1 bound at its target promoters, which ensures continued *che-1* expression by preferentially binding the *che-1* promoter. We provide experimental evidence for this mechanism by showing that *che-1* expression was resilient to induced transient CHE-1 depletion, while both expression of CHE-1 targets and ASE function were lost. We identified a 130 bp *che-1* promoter fragment responsible for this resilience, with deletion of a homeodomain binding site in this fragment causing stochastic loss of ASE identity long after its determination. Because network architectures that support this mechanism are highly conserved in cell differentiation, it may explain stable cell fate maintenance in many systems.

## Editor's evaluation

This paper will be of interest to developmental biologists and neurobiologists who study the molecular mechanisms underlying induction and maintenance of cell fate. A combination of cutting-edge molecular genetic approaches in *C. elegans* together with mathematical modeling suggests an interesting mechanism for life-long maintenance of neuronal identity and function.

## Introduction

In most animal tissues, terminally differentiated cells are renewed on timescales of days to months (*Leblond and Walker, 1956*), yet the nervous system is unique, as most neurons hardly renew at all (*Ming and Song, 2005*). Many mature neuron types exist that differ in the expression of hundreds of type-specific genes (*Deneris and Hobert, 2014*). How neuronal cells maintain this terminally differentiated state over such long timescales – decades, in the case of humans – is not understood.

Differentiation of neuron types is often controlled by a small subset of transcription factors, sometimes even a single one, that are called 'terminal selectors' (*Hobert, 2016*; *Hobert and Kratsios, 2019*). These act through a conserved network motif called a single-input module (*Alon, 2007*): they bind to specific *cis*-regulatory control elements to induce both their own expression and that of the downstream target genes that define the neuronal type (*Figure 1A*). Such terminal selector networks have been found to underlie differentiation of several neuron types in the nematode *Caenorhabditis elegans* (*Deneris and Hobert, 2014*; *Hobert, 2016*), photoreceptor subtypes in *Drosophila* (*Hsiao*

**Figure 1.** Loss of ASE neuron fate upon transient CHE-1 depletion. (**A**) The terminal selector gene *che-1* induces its own expression and that of 500–1000 target genes that together determine ASE neuron fate. Positive autoregulation of *che-1* expression could result in bistable, switch-like behavior. (**B**) Bistability generates sustained terminal selector expression upon transient induction (1), that is resilient to short periods of terminal selector depletion (2). However, bistable switches remain reversible and will lose terminal selector expression upon sufficiently long depletion (3), while irreversible switches will always recover. (**C**) Transient CHE-1 depletion using Auxin-Induced Degradation (AID). *che-1::GFP::AID* L1 larvae (CHE-1::GFP::AID) or young adults (chemotaxis) were exposed for different time periods to 1 mM auxin to induce CHE-1::GFP:AID degradation, and were subsequently characterized after a 24- or 48-hr recovery period. (**D**) CHE-1::GFP::AID fluorescence in *che-1::GFP::AID* animals before (left) and after 24 hr auxin treatment (middle), and after a subsequent 24 hr recovery off auxin (right). Even though CHE-1::GFP::AID is lost from ASE neurons after auxin treatment, it reappears after recovery off auxin. (**E**) Response to 10 mM NaCl for wild-type animals, *che-1(p679)* mutants defective in NaCl chemotaxis, and *che-1::GFP::AID* animals exposed to auxin for 24–96 hr (24A – 96A) tested directly (0 R) or after 48 hr recovery (48 R). *che-1::GFP::AID* animals on auxin showed a chemotaxis defect similar to *che-1(p679)* mutants. *che-1::GFP::AID* animals recovered chemotaxis to NaCl after 24 or 48 hr on auxin, but exhibited a persistent chemotaxis defect after sufficiently long, transient CHE-1::GFP::AID depletion. (**F**) Fraction of animals that recovered CHE-1::GFP::AID expression 48 hr after auxin treatment of increasing length. No animals recovered CHE-1::GFP::AID expression after 120 hr depletion. Error bars in (**E**) and (**F**) represent mean of four assays ± S.E.M. *$p < 0.05$, **$p < 0.01$, ***$p < 0.001$, n.s. indicates not significant. In (**E**) significance is compared to *che-1(p679)* mutants (black) or *che-1::GFP::AID* animals without auxin (red).

The online version of this article includes the following source data and figure supplement(s) for figure 1:

**Source data 1.** Data and scripts for *Figure 1* and related figure supplements.

*Figure 1 continued on next page*

*Figure 1 continued*

**Figure supplement 1.** CHE-1 recovery after depletion.

**Figure supplement 2.** CHE-1 depletion and NaCl chemotaxis.

*et al., 2013*) and dopaminergic neurons in mice (*Ninkovic et al., 2010*), indicating that they form an evolutionary conserved principle for neuron type determination.

Terminal selectors positively regulate their own expression, raising the possibility that they act as bistable genetic switches. Such switches are widespread in biology (*Alon, 2007*; *Ferrell, 2002*) and are often seen as an attractive mechanism to explain cell fate determination. In this hypothesis, at the time of determination, transient signals induce expression of terminal selectors, which then maintain their own expression, and that of all target genes, by autoregulation in the subsequent absence of these signals (*Figure 1B*; *Hobert, 2008*). However, a key weakness of bistable switches is that they remain reversible at all times, with a transient decrease in terminal selector levels potentially sufficient to lose terminal selector expression and, presumably, cell fate (*Figure 1B*). Indeed, bistable genetic switches often suffer from stochastic transitions between their different states due to molecular noise, that is random fluctuations in the levels of their core components (*Acar et al., 2005*; *Axelrod et al., 2015*; *Gupta et al., 2011*; *Nevozhay et al., 2012*; *Ozbudak et al., 2004*). Hence, it is often assumed that stable cell fate maintenance must require additional feedback mechanisms, such as histone or chromatin modifications (*Orlando, 2003*; *Ringrose and Paro, 2007*), that lock-in cell fate in an irreversible manner.

Here, we studied long-term maintenance of the salt-sensing ASE neuron type in the nematode *C. elegans* (*Figure 1A*). ASE type is controlled by the terminal selector CHE-1, a transcription factor whose expression is transiently induced by the nuclear hormone receptor NHR-67 at the time of determination (*Sarin et al., 2009*). CHE-1 induces the expression of 500–1000 ASE-specific target genes, such as chemosensory receptors, ion-channels, and neuropeptides, by binding ASE motifs within their promoters (*Etchberger et al., 2007*). Its continued presence is required for expression of target genes after neuron type determination (*Etchberger et al., 2009*). CHE-1 also upregulates its own expression. This positive feedback loop is necessary for maintaining *che-1* expression and ASE cell fate directly after cell fate determination (*Etchberger et al., 2007*; *Leyva-Díaz and Hobert, 2019*). However, whether this positive feedback loop by itself is sufficient to ensure life-long maintenance of ASE fate is unknown. The impact of molecular noise, such as variability in CHE-1 protein copy number, on ASE fate maintenance has not been studied. Overall, it is an open question how a reversible, bistable switch based on positive CHE-1 autoregulation would remain sufficiently stable for the animal's lifetime to maintain ASE fate, or if additional mechanisms are necessary to ensure its stability.

Here, we show that sufficiently long, transiently induced depletion of CHE-1 causes permanent loss of ASE fate, indicating that it is controlled by a switch that remains reversible long after specification. This raises the question how the switch is protected against molecular noise, which could cause it to spontaneously lose ASE fate. Combining experimental measurements of the key parameters that control the magnitude of noise, that is the copy numbers and lifetimes of *che-1* mRNA and protein, with stochastic models of the *che-1* genetic network, revealed a novel mechanism, 'target reservoir buffering', that dramatically increased switch stability. Our simulations revealed that this stability resulted from the presence of a reservoir of CHE-1 protein bound at the promoters of its target genes, coupled with preferential binding of CHE-1 to the *che-1* promoter compared to the promoters of its other targets. This led to exceedingly stable ON states (high *che-1* expression), with spontaneous transitions to the OFF state (no *che-1* expression) observed at rates of $< 10^{-3}$ /year. Consistent with this mechanism, we observed that upon induced CHE-1 depletion in vivo, *che-1* mRNA expression remained present, even when expression of other target genes vanished together with the animal's ability to respond to salt. This was followed by full recovery of CHE-1 protein levels, target gene expression and chemosensation if induced CHE-1 depletion was sufficiently short. We found a 130 bp promoter region surrounding the *che-1* ASE motif responsible for this resilience of *che-1* expression to CHE-1 depletion. This region contained a homeodomain protein binding site that, when mutated, caused stochastic loss of *che-1* expression and ASE function well after ASE specification, indicating a strong decrease in stability of the ON state. We therefore speculate that homeodomain proteins have a role in maintaining and stabilizing ASE fate, potentially by recruiting CHE-1 preferentially to its own promoter.

# Results

## Loss of ASE neuron fate upon transient CHE-1 depletion

To test whether positive autoregulation of *che-1* expression is necessary for ASE fate maintenance, we depleted CHE-1 protein levels in ASE neurons in vivo, using the auxin-inducible degradation system (*Serrano-Saiz et al., 2018*; *Zhang et al., 2015*). *che-1::GFP::AID* animals were exposed to 1 mM auxin to induce CHE-1::GFP::AID depletion (*Figure 1C*). CHE-1::GFP::AID was strongly reduced after ~1.5 hr, and undetectable after ~3 hr exposure to auxin (*Figure 1D*, *Figure 1—figure supplement 1A*). To examine how CHE-1::GFP::AID depletion impacted ASE function, we used a quadrant chemotaxis assay to quantify the chemotaxis response of CHE-1::GFP::AID depleted animals (*Jansen et al., 2002*; *Wicks et al., 2000*). In this assay, animals choose between two agar quadrants with NaCl and two without (*Figure 1—figure supplement 2A*). *che-1::GFP::AID* animals exposed to auxin for 24 hr showed reduced chemotaxis to NaCl (*P* < 0.001) (*Figure 1E*, *Figure 1—figure supplement 2B*). This agrees with previous results showing that permanent inhibition of *che-1* expression by RNAi in larvae resulted in reduced expression of CHE-1 target genes (*Etchberger et al., 2009*). However, by inhibiting *che-1* expression permanently, rather than transiently, these results left open the possibility that, after initial ASE determination, *che-1* expression is maintained independently of CHE-1 in an irreversible manner.

In contrast, if *che-1* expression is bistable and reversible, then a sufficiently long period of transient CHE-1 depletion should result in permanent loss of *che-1* expression (*Figure 1B*). To test this, we exposed animals to auxin for increasing time intervals (*Figure 1C*), and analysed CHE-1::GFP::AID expression and NaCl chemotaxis after 24 or 48 hr recovery on plates without auxin (*Figure 1D–F*, *Figure 1—figure supplement 1B, C*, *Figure 1—figure supplement 2C*). In animals exposed to auxin for 24 hr, both CHE-1::GFP::AID expression and NaCl chemotaxis returned to wild-type levels after 24 hr recovery (*Figure 1D*, *Figure 1—figure supplement 1B*, *Figure 1—figure supplement 2C*). However, after 48 hr auxin exposure, CHE-1::GFP::AID did not return in 8/29 and 3/12 animals after 24 and 48 hr recovery, respectively (*Figure 1F*, *Figure 1—figure supplement 1B*). This fraction further increased with the duration of auxin exposure, with none of the animals recovering CHE-1::GFP::AID expression after 120 hr on auxin (*Figure 1F*). Similarly, the chemotaxis response did not recover in animals exposed to auxin for 96 hr or longer (*Figure 1E*, *Figure 1—figure supplement 2C*, *Supplementary file 1*), suggesting that the absence of CHE-1::GFP::AID resulted in loss of ASE identity.

The long duration of auxin exposure meant that animals aged considerably during the experiments, raising the question whether the failure to recover from CHE-1 depletion reflected a general deterioration of ASE function with age. However, untreated animals did not show a significant decrease in chemotaxis with age (up to 168 hr after hatching, *Figure 1—figure supplement 2D*). Moreover, 72 hr old *che-1::GFP::AID* animals, exposed to auxin for 48 hr, were able to recover NaCl chemotaxis after 24 and 48 hr off auxin (*Figure 1—figure supplement 2E*), indicating that ASE neurons are functional in old animals and are capable of recovering from CHE-1 depletion. In mice, the removal of a terminal selector protein in adult neurons can result in cell death (*Serrano-Saiz et al., 2018*), implying that the lack of CHE-1::GFP::AID recovery could be due to ASE cell death, rather than a switch to the OFF state. We therefore performed CHE-1::GFP::AID depletion in animals carrying an *osm-3::GFP* reporter (*Figure 1—figure supplement 1D*). OSM-3 functions in ciliated neurons and is expressed in 10 pairs of amphid neurons (AWB, AWC, ASE, ASG, ASH, ASI, ASJ, ASK, ADF, ADL) (*Evans et al., 2006*; *Nokes et al., 2009*; *Tabish et al., 1995*). In contrast to existing ASE-specific transcriptional reporters, we expected that *osm-3* expression was not controlled by CHE-1 and would therefore remain expressed upon CHE-1 depletion. Indeed, when we exposed 24 hr old *che-1::GFP::AID; osm-3::GFP* animals to auxin for 108 hr, we counted 10 GFP-positive neurons on one side of the animal (10.1 ± 0.9, n = 26), similar to the number of neurons in control animals of the same age but not exposed to auxin (9.9 ± 0.7, n = 30), indicating that ASE neurons did not die even after prolonged CHE-1::GFP::AID depletion. Overall, these observations indicated that CHE-1 controls ASE fate as a bistable switch that, even though it can withstand strong decreases in CHE-1 level for a limited time, remains fully reversible long after its induction.

## Copy number and lifetime of *che-1* mRNA and protein

The observed reversibility of the ON state raises the question how spontaneous transitions to the OFF state, due to random fluctuations in CHE-1 level, are prevented under normal conditions. Theoretical

studies of genetic switches suggest that the probability of such transitions decreases with increasing average copy number and lifetime of the transcription factors involved (*Mehta et al., 2008*; *Walczak et al., 2005*; *Warren and ten Wolde, 2005*). Therefore, we determined copy numbers and lifetimes of *che-1* mRNA and protein.

First, we measured the absolute number of *che-1* mRNA and protein molecules in ASE neurons. We used single-molecule FISH (smFISH) (*Ji and van Oudenaarden, 2012*) to count individual *che-1* mRNA molecules. As a benchmark, we also measured mRNA levels of the putative NaCl receptors *gcy-14* and *gcy-22*, and the ion channel subunits *del-2* and *tax-2*, which are CHE-1 target genes with >1 ASE motif in their promoter region (*Coburn and Bargmann, 1996*; *Ortiz et al., 2009*). In embryos, we found that *che-1* mRNA levels peaked at 26 ± 6 mRNAs/cell during the bean stage, that is the time of ASE neuron determination (*Figure 2A and B*), and fell to 6 ± 3 mRNAs/cell from the comma stage onwards. In contrast, the CHE-1 target gene *gcy-22* showed a steady increase in mRNA levels during development and surpassed *che-1* mRNA expression after the 1.5-fold stage. In larvae, we also found low *che-1* mRNA levels, with 5 ± 2 mRNAs in the left (ASEL) and 7 ± 3 mRNAs in the right ASE neuron (ASER). *che-1* expression was low compared to the panel of CHE-1 target genes examined (*Figure 2C and D*), with *del-2*, *gcy-14*, and *gcy-22* expressed at significantly higher levels. Moreover, *che-1* mRNA levels remained low during all four larval stages, L1-L4, with slightly higher mRNA levels in the ASER, compared to the ASEL neuron (*Figure 2E*). For this entire period, the target genes with the highest expression, *gcy-14* and *gcy-22*, remained expressed well above *che-1* levels. Overall, these results show that *che-1* expression is biphasic: an initiation phase with high *che-1* expression level when ASE fate is induced in the embryo, followed by a maintenance phase with *che-1* expressed at a low level where molecular copy number fluctuations likely have significant impact.

We also estimated the absolute CHE-1 protein copy number in the ASE neurons. We immersed endogenously tagged *che-1::GFP* animals (*Leyva-Díaz and Hobert, 2019*) in eGFP, and calculated the CHE-1::GFP protein number by comparing the CHE-1::GFP fluorescence inside the ASE neurons with the ambient fluorescence of eGFP (*Figure 2F*; *Gregor et al., 2007*). For all stages of post-embryonic development, we found an average CHE-1::GFP copy number of 900 ± 250 CHE-1::GFP (~325 nM) and 600 ± 120 CHE-1::GFP (~260 nM) molecules/cell for the ASER and ASEL neuron, respectively (*Figure 2F*, *Figure 2—figure supplement 1A*), with slightly lower copy numbers observed in the embryo (*Figure 2—figure supplement 1B*). Overall, the observed range of CHE-1::GFP protein levels, 500–1400 molecules/cell, is comparable to the number of CHE-1 binding sites, 500–1000, in the promoters of CHE-1 target genes, as estimated by the number of ASE motifs detected previously (*Etchberger et al., 2007*) and by our analysis (Materials and methods), indicating that CHE-1 targets compete for a limited pool of CHE-1 protein.

Next, we measured *che-1* mRNA and protein lifetimes. To determine *che-1* mRNA lifetimes in ASE neurons, we transiently overexpressed *che-1* mRNA, using a *hsp16.41p::che-1* heat-shock-inducible construct (*Patel and Hobert, 2017*). By exposing animals to 37°C for ~10 min, we raised *che-1* mRNA levels in the ASE neurons 5-fold, to 24 ± 4 mRNAs/cell ~10 min after heat shock (*Figure 3A and B*). Next, we shifted animals to 20°C to return *che-1* expression to its pre-induction level and fixed animals at ~17 min intervals to quantify *che-1* mRNA levels using smFISH. We found that *che-1* mRNA levels returned to pre-induction values after ~60 min. By fitting the measured *che-1* mRNA levels to an exponential function, we obtained a *che-1* mRNA half-life of 17 ± 4 min. To measure CHE-1 protein lifetime, we used Fluorescence Recovery after Photobleaching (FRAP) in *che-1::GFP* animals. CHE-1::GFP was bleached to approximately ~20% of the original fluorescence level and we measured the recovery of the CHE-1::GFP signal over time, until it reached pre-bleaching levels, which occurred within ~3 hrs (*Figure 3C and D*). To estimate CHE-1::GFP protein lifetime, we fitted an exponential recovery curve to the experimental data for each individual animal, resulting in an average measured half-life of 83 ± 20 min.

The CHE-1 protein half-life we measured was short compared to the general protein half-lives reported in *C. elegans* adults (*Dhondt et al., 2017*). Instead, the measured *che-1* mRNA and protein half-lives were comparable to typical values reported for transcription factors in mammalian systems (*Fornasiero et al., 2018*; *Schofield et al., 2018*; *Yang et al., 2003*), indicating that CHE-1 lifetime was not increased as a strategy to ensure stability of ASE fate maintenance. In fact, *che-1* mRNA and protein turnover is rapid compared to the 2–3 week *C. elegans* lifespan over which ASE identity must be maintained.

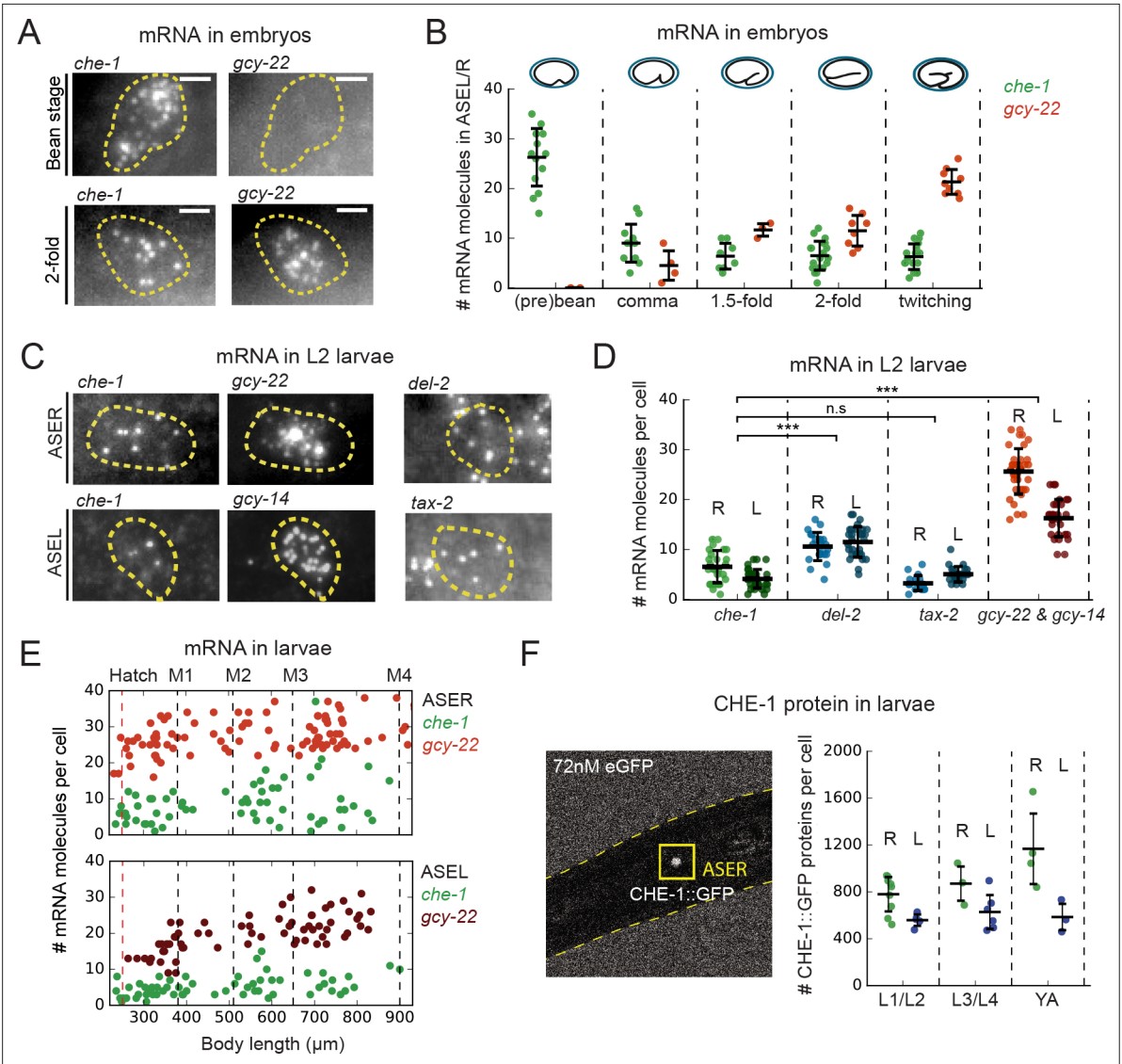

**Figure 2.** *che-1* mRNA and protein copy numbers in ASE neurons. (**A**) Expression of *che-1* and the CHE-1 target *gcy-22* in embryos at the bean (top) and 2-fold stage (bottom). Each spot is a single mRNA molecule visualized by single molecule FISH (smFISH). Dashed lines outline ASE neuron cell bodies. Scale bar: 2 µm. (**B**) *che-1* (green) and *gcy-22* (red) mRNA levels during embryonic development. *che-1* expression peaks during specification, but falls as expression of CHE-1 target genes rise. (**C**) Expression of *che-1* and CHE-1 targets *gcy-22*, *gcy-14*, *tax-2* and *del-2* visualized by smFISH in L2 larvae. Dashed lines outline left and right ASE neurons (ASEL/R). Scalebar: 2 µm. (**D**) Quantification of expression of *che-1* and CHE-1 targets in ASEL (L) and ASER (R) in L2 larvae. *che-1* mRNA levels are low compared to other CHE-1 target genes. (**E**) Low *che-1* expression (green, ASER/L) compared to expression of CHE-1 targets *gcy-22* (red, ASER) and *gcy-14* (red, ASEL) throughout larval development. Body length corresponds to developmental time, with approximate timing of hatching and molts between larval stages L1-L4 indicated by vertical lines. (**F**) Left panel: two-photon microscopy image of L2 larva expressing endogenously-tagged CHE-1::GFP, immersed in 72 nM eGFP for calibration. Right panel: CHE-1::GFP protein molecules in ASER (green), (R) and ASEL (blue), (L) at different stages of post-embryonic development. The number of CHE-1 proteins is comparable to the predicted number of CHE-1-binding sites. Error bars in B, D, and F represent mean ± SD. ***p < 0.001.

The online version of this article includes the following source data and figure supplement(s) for figure 2:

**Source data 1.** Data and scripts for *Figure 2* and related figure supplements.

**Figure supplement 1.** CHE-1 copy numbers in larvae and embryos.

## Stochastic simulations identify stable cell fate maintenance parameters

The measurements of *che-1* mRNA and protein copy numbers and lifetimes allowed us to perform realistic simulations of the CHE-1 switch to estimate its stability against stochastic fluctuations. We constructed stochastic models that included production and decay of *che-1* mRNA and protein

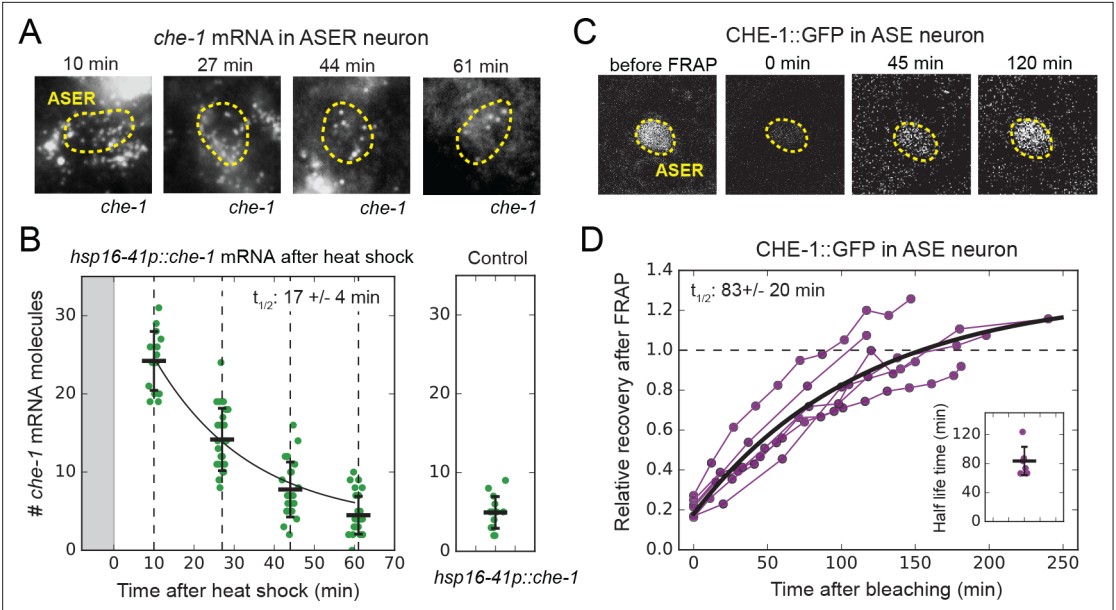

**Figure 3.** *che-1* mRNA and protein lifetimes. (**A**) *che-1* mRNAs in L2 larvae at different times after induction of *che-1* by a 37 °C heat shock in *hsp16-41p::che-1* animals, visualized by smFISH. Dashed lines outline ASER neuron. (**B**) *che-1* mRNA level in ASE neurons of individual L2 animals (green) as function of time after a 10 min heat shock (gray area). Black line is the fitted decay curve. Control L2 larvae did not receive a heat shock. The measured *che-1* mRNA half-life was 17 ± 4 mins. (**C**) CHE-1::GFP fluorescence recovery after photobleaching (FRAP) in the ASER neuron of a single L4 animal. Time is indicated relative to bleaching of CHE-1::GFP. (**D**) Fluorescence recovery of CHE-1::GFP in ASE neurons of L4 or young adult animals (n = 6). An exponential recovery curve model was fitted to data of each individual animal (black line indicates the average recovery curve). The inset shows the fitted half-life for each individual animal. The average measured CHE-1::GFP protein half-life was 83 ± 20 min. Error bars in B represent mean ± SD. n ≥ 10 in B and n = 6 in D.

The online version of this article includes the following source data for figure 3:

**Source data 1.** Data and scripts for *Figure 3* and related figure supplements.

molecules, binding of CHE-1 to the promoter of *che-1* and target genes, and target gene expression (*Figure 4A*). We examined two bistable models that differed in binding of CHE-1 to its own promoter. In the first model, we assume that CHE-1 binds as a monomer to induce *che-1* expression. This model agrees with the observation that the *che-1* promoter contains only a single ASE motif (*Etchberger et al., 2007*), but lacks cooperativity in *che-1* induction. Because cooperativity is considered important for generating bistability (*Ferrell, 2002*), we also included a second model where *che-1* induction is cooperative, by assuming that expression occurs only when two CHE-1 molecules bind the *che-1* promoter.

The two models have 8 and 9 parameters, respectively, of which experimental data fixed 6 (Materials and methods). The production and degradation rates of *che-1* mRNA ($f_m$, $b_m$) and protein ($f_c$, $b_c$) were fully determined by the measured copy numbers and lifetimes. For the CHE-1 binding rate ($f_O$), we assumed the diffusion-limited rate, that is, the highest physically possible binding rate. Based both on previous analysis (*Etchberger et al., 2007*) and our own, we examined model dynamics for $N_T$ = 500 or 1000 CHE-1 target genes, that is, either smaller or larger than the mean number of CHE-1 proteins (900 molecules/cell). The only free parameters were dissociation rates of CHE-1 from its own promoter ($b_O$ or $b_1$, $b_2$) and from the other targets ($b_T$). We varied these between 0.1 and 100 $s^{-1}$, corresponding to dissociation constants of ~1-$10^3$ nM and consistent with values measured for transcription factors (*Gebhardt et al., 2013*; *Jung et al., 2018*). We simulated the models using the Gillespie algorithm (*Gillespie, 2002*). In general, our stochastic simulations showed that molecular noise was sufficiently strong to induce spontaneous transitions from the ON to the OFF state on the timescale of hours or days, indicating that the measured copy numbers and lifetimes by themselves were not sufficient to generate stability against fluctuations. However, we also identified parameter combinations for which the CHE-1 switch remained in the ON state for at least a week (*Figure 4B*).

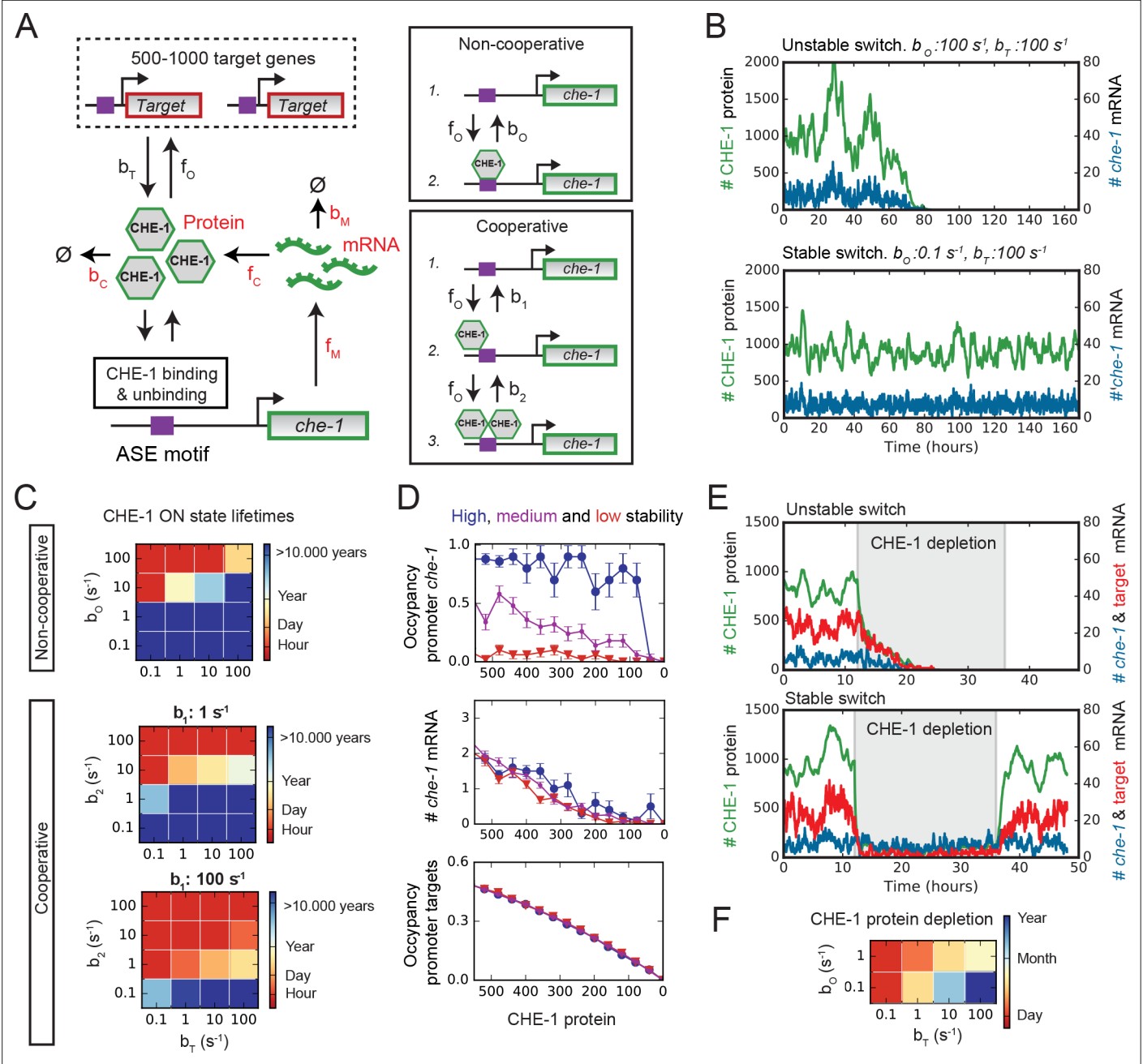

**Figure 4.** Stable ON state by preferential binding of CHE-1 to its own promoter. (**A**) Overview of the bistable, stochastic CHE-1 switch model, including production and degradation of *che-1* mRNA and protein, and binding of CHE-1 protein to its own promoter and other target genes. Parameters constrained by experiments are red. Inset: CHE-1 binding is modelled as monomers (non-cooperative) or dimers (cooperative). (**B**) Stochastic simulations of the non-cooperative model for parameters with an unstable (top) or stable (bottom) ON state (*che-1* expression), showing levels of *che-1* mRNA (blue) and protein (green). For parameters resulting in an unstable switch, stochastic fluctuations induce a spontaneous transition to the OFF state (no *che-1* expression). (**C**) Average ON state lifetimes calculated using Forward Flux Sampling (FFS) as function of CHE-1 dissociation rates from its own promoter ($b_O$ or $b_1, b_2$) and its target genes ($b_T$) for the non-cooperative and cooperative model. Stable ON state occurs for high *che-1* promoter occupancy by CHE-1 ($b_O$ <1 or $b_1, b_2$ <1) and preferential affinity of CHE-1 for its own promoter compared to that of its target genes ($b_O$ or $b_2 \ll b_T$). (**D**) Average CHE-1 occupancy of the promoter of *che-1* (top) and other target genes (bottom), and average *che-1* mRNA level (middle) during spontaneous transitions from the ON to the OFF state, as sampled by FFS. Shown are transition paths for parameters with low (red, $b_O$=100 $s^{-1}$), medium (magenta, $b_O$=10 $s^{-1}$), and high (blue, $b_O$=1 $s^{-1}$) stability of the ON state, with $b_T$=10 $s^{-1}$. For simulations with a stable ON state, the *che-1* promoter remained fully occupied by CHE-1, even as CHE-1 protein levels approached zero, in contrast to the occupancy of promoters of other CHE-1 target genes. (**E**) Simulations showing the impact of transient depletion of CHE-1 protein (green) on mRNA levels of *che-1* (blue) and a target gene (red). CHE-1 is depleted to 100 molecules/cell by a transient increase in degradation ($b_C$; grey region). For parameters with an unstable ON state (top), both

*Figure 4 continued on next page*

Figure 4 continued

*che-1* and target gene mRNA levels fall rapidly, and do not recover when CHE-1 depletion ceases. For a stable ON state (bottom), expression of *che-1* is unaffected by CHE-1 depletion, leading to full recovery once CHE-1 depletion ends. (**F**) Average ON state lifetimes, calculated by FFS, during CHE-1 depletion to 100 molecules/cell. Parameter combinations with a stable ON state under normal conditions maintain *che-1* expression for hours or days under induced CHE-1 depletion.

The online version of this article includes the following source data and figure supplement(s) for figure 4:

**Source data 1.** Data and scripts for *Figure 4* and related figure supplements.

**Figure supplement 1.** Dependence of ON state stability on number of targets and cooperativity.

For simulations with high switch stability, brute-force Gillespie simulations were too computationally demanding to directly measure the ON state lifetime. Instead, we used Forward Flux Sampling (FFS), a computational method to efficiently sample rare transition paths between states in multi-stable systems (*Allen et al., 2006*). Using this approach, we observed parameter combinations with very high stability, that is lifetimes of many years, independent of the degree of cooperativity or number of target genes (*Figure 4C*, *Figure 4—figure supplement 1A,B*). In general, we observed increasing lifetimes for decreasing $b_O$ or $b_1, b_2$, that is, higher occupancy of the *che-1* promoter by CHE-1. Moreover, the longest lifetimes were found when $b_O, b_2 \ll b_T$, that is, when CHE-1 had a much higher affinity for its own promoter than for its other targets. In this regime, we observed average lifetimes of >1 years. Note, however, that despite the long lifetimes in this regime, spontaneous transitions to the OFF state are rapid and occur as a random Poisson process, with a transition possible at any time, albeit with low probability. For such a Poisson process, ON state lifetimes of years are required for the probability of spontaneous loss of ASE fate during their ~2-week lifetime to be less than $10^{-6}$ (Materials and methods), the frequency of spontaneous mutations per gene per generation in *C. elegans* (*Anderson, 1995*).

## Stability against stochastic fluctuations by preferential binding of CHE-1 to its own promoter

We found that high stability of the ON state required $b_O, b_2 \ll b_T$, that is, CHE-1 binds its own promoter much more strongly than that of its other targets. An explanation for this emerged when we compared transition paths for spontaneous transitions to the OFF state, between parameter combinations that exhibited high (>1 years lifetime), medium (~12 days lifetime), and low (~5 hrs lifetime) stability (*Figure 4D*, *Figure 4—figure supplement 1C*). For parameters with low stability, we found that, as CHE-1 protein levels fell during spontaneous transitions to the OFF state, both the average *che-1* mRNA number and the fraction of CHE-1 target promoters occupied by CHE-1 decreased, with very low occupancy even of the *che-1* promoter itself close to the end of the transition. In contrast, for parameters with high stability, we found that *che-1* promoter occupancy was high, and the *che-1* promoter remained bound by CHE-1 until the end of the transition, whereas CHE-1 binding was lost earlier on other promoters. These results suggested that high stability arises as a result of a strong preference for CHE-1 protein to bind to the *che-1* promoter, thereby making *che-1* expression insensitive to strong, stochastic decreases in CHE-1 level.

To test this idea, we ran simulations that included a transient, 24 hr depletion of CHE-1, implemented by a temporary increase in the CHE-1 protein degradation rate that reduced CHE-1–100 molecules/cell (*Figure 4E*). Indeed, we found that simulations with unstable switches, that is where $b_O, b_2 > b_T$, were highly sensitive to such depletions, with *che-1* mRNA rapidly falling to such low levels that CHE-1 protein levels and, hence, the ON state, were not recovered when CHE-1 depletion ceased. In contrast, even though the mRNA levels of target genes fell rapidly, simulations with highly stable switches maintained normal *che-1* mRNA levels for many days (*Figure 4F*, *Figure 4—figure supplement 1D*), allowing the system to successfully recover CHE-1 protein levels and the ON state if CHE-1 depletion was removed sufficiently rapidly. Finally, for longer, 48 hr depletion, a fraction of simulations failed to recover CHE-1 levels, as observed experimentally (*Figure 4—figure supplement 1E*, *Figure 1F*). In these simulations, lack of recovery was due to stochastic loss of all CHE-1 proteins during the period of induced depletion.

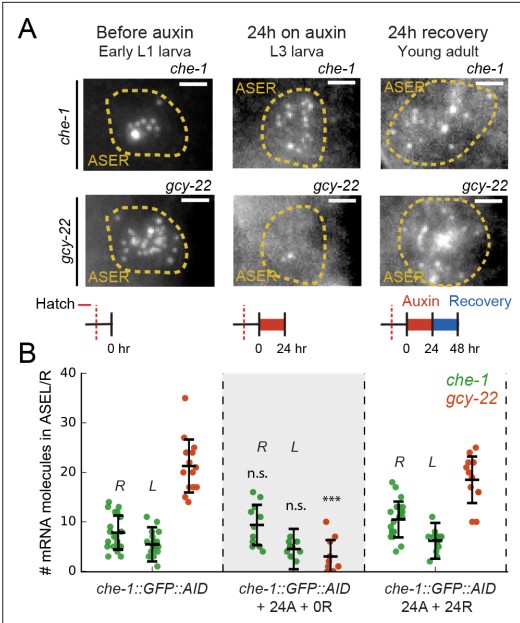

**Figure 5.** Maintenance of *che-1* expression during transient CHE-1 depletion. (**A**) *che-1* and *gcy-22* mRNA levels in ASER neurons, visualized by smFISH in *che-1::GFP::AID* animals, at different moments in a 24 hr auxin and 24 hr recovery treatment. Schematics indicate the time and duration of auxin treatment and recovery. Scale bar: 2 µm. (**B**) *che-1::GFP::AID* (green, ASEL/R) and *gcy-22* (red, ASER) mRNA levels quantified in *che-1::GFP::AID* animals at different times in a 24 hr auxin and 24 hr recovery treatment. Upon depletion of CHE-1::GFP::AID protein, *che-1* expression was not impacted, while *gcy-22* levels strongly decreased. *gcy-22* expression rose to wild-type levels after 24 hr recovery off auxin. Error bars in B represent mean ± SD. n ≥ 10. ***p < 0.001.

The online version of this article includes the following source data for figure 5:

**Source data 1.** Data and scripts for *Figure 5* and related figure supplements.

## In vivo CHE-1 depletion decreases target gene but not *che-1* expression

Our simulation results were similar to our experimental observation that most animals fully regain CHE-1::GFP::AID levels even after 24–48 hr of induced CHE-1::GFP::AID depletion (*Figure 1C–F*, *Figure 1—figure supplement 1B,C*). To test whether this reflected insensitivity of *che-1* expression to low CHE-1 protein levels (*Figure 4E and F*), we used smFISH to compare the impact of auxin-mediated CHE-1::GFP::AID depletion on the mRNA levels of *che-1::GFP::AID* and other target genes, focusing on *gcy-22* as the most highly expressed in our panel (*Figure 5A and B*). Indeed, *gcy-22* mRNA levels were very low in most *che-1::GFP::AID* animals after 24 hr on auxin. In striking contrast, *che-1::GFP::AID* mRNA levels were close to wild-type. As auxin is dissolved in EtOH, these experiments were performed at 0.25% EtOH, which by itself could impact gene expression. However, we found that mRNA levels were not impacted by the presence of 0.25% EtOH (*Figure 2—figure supplement 1C*), indicating that low *gcy-22* levels resulted from CHE-1::GFP::AID depletion. After 24 hr without auxin, *gcy-22* mRNA levels had increased significantly (*Figure 5A and B*), consistent with the recovery of CHE-1::GFP::AID levels and chemotaxis to NaCl in these animals (*Figure 1D and E*). Overall, these results were in full agreement with our model predictions but raised the question what properties of the *che-1* promoter were responsible for its resilience to CHE-1 depletion.

## Sequences flanking the *che-1* ASE motif are required for *che-1* expression during CHE-1 depletion

Previous studies identified a single ASE motif 242 bp upstream of the *che-1* ATG start codon as required for autoregulation of *che-1* expression (*Etchberger et al., 2007*; *Leyva-Díaz and Hobert, 2019*). This ASE motif differs in 6 bp from the sequence in the *gcy-22* promoter (*Figure 6A*), which might explain the divergent effects of CHE-1 depletion on *che-1* and *gcy-22* expression. To test this hypothesis, we used CRISPR/Cas9 in *che-1::GFP::AID* animals to replace the 12 bp ASE motif in the *che-1* promoter with that of *gcy-22*, and vice versa (*Figure 6A*). The resulting mutant animals showed wild-type chemotaxis to NaCl and exhibited *che-1::GFP::AID* and *gcy-22* mRNA levels similar to wild-type (*Figure 6B and F*, *Figure 6—figure supplement 1A*), indicating that replacing ASE motifs did not impact ASE specification and target gene expression. Moreover, when we depleted CHE-1::GFP::AID protein using auxin, *gcy-22* expression in (ASE*che-1*)p::*gcy-22* animals almost completely vanished (*Figure 6F*), as in animals with a wild-type *gcy-22* promoter. Overall, the ASE motif itself could not explain the observed differences in *che-1* and *gcy-22* expression. This agrees with previous results that showed a similar calculated affinity score of CHE-1 for the *che-1* and *gcy-22* ASE motif, despite sequence differences (*Etchberger et al., 2009*).

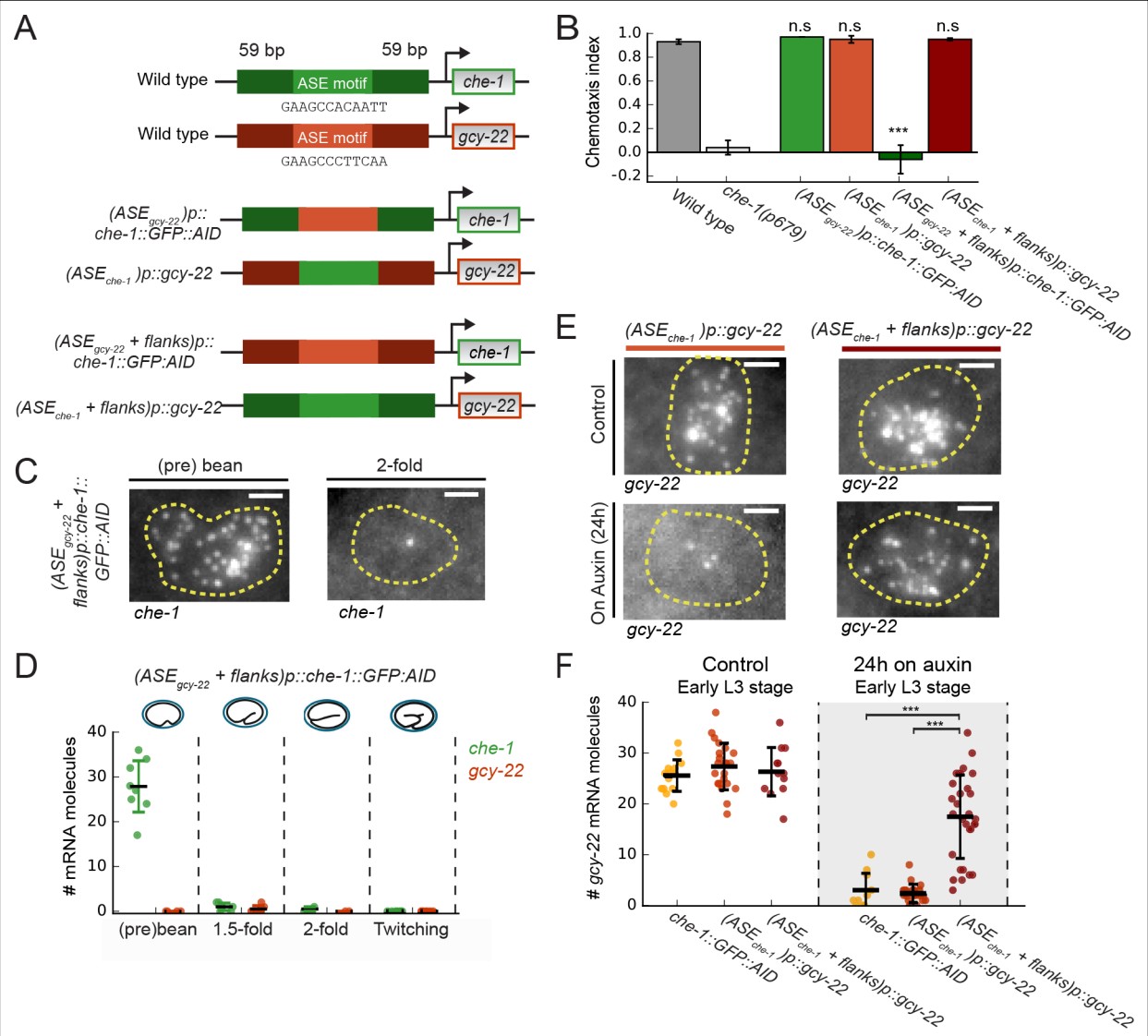

**Figure 6.** Region flanking CHE-1-binding site ensures resilience to CHE-1 depletion. (**A**) Overview of *che-1* promoter mutants generated in the *che-1::GFP:AID* background. We exchanged either the *che-1* (green) or *gcy-22* (red) ASE motif, which binds CHE-1, or a larger region that includes 59 bp flanks on either side. (**B**) Average chemotaxis index for response to 10 mM NaCl, of wild-type and *che-1(p679)* animals, and *che-1* promoter mutants. Exchange of ASE motifs between *che-1* and *gcy-22* promoters did not affect chemotaxis. Replacing the *che-1* ASE motif with flanks for that of *gcy-22* abolished chemotaxis to NaCl; the reverse had no effect. (**C**) *che-1* expression visualized by smFISH in ASE neurons of (*ASE_{gcy-22}*+ *flanks*)*p::che-1::GFP::AID* embryos. In twofold embryos, *che-1::GFP::AID* mRNA levels were low. Scale bar: 1.5 μm. (**D**) *che-1::GFP::AID* and *gcy-22* mRNA levels quantified in the ASE neurons of (*ASE_{gcy-22}*+ *flanks*)*p::che-1::GFP::AID* embryos. After initial high *che-1::GFP::AID* expression at the time of ASE specification, *che-1::GFP::AID* and *gcy-22* expression was almost absent, indicating a failure of ASE fate maintenance. (**E**) *gcy-22* expression under normal conditions or upon CHE-1::GFP::AID depletion by auxin, in L3 larvae of *che-1* promoter mutants. Scale bar: 1.5 μm. (**F**) Quantification of *gcy-22* mRNA levels upon auxin-induced CHE-1::GFP::AID depletion, in *gcy-22* promoter mutants. In (*ASE_{gcy-22}*)*p::che-1::GFP::AID* animals, *gcy-22* mRNA levels fell on auxin, as observed before. However, in (*ASE_{che-1}*+ *flanks*)*p::gcy-22* animals treated with auxin *gcy-22* levels remained high. Thus, the region flanking the *che-1* ASE motif drives the maintenance of *che-1* expression during CHE-1 protein depletion. Error bars in B represent mean ± S.E.M, D and F represent mean ± SD. n ≥ 10. ***p < 0.001.

The online version of this article includes the following source data and figure supplement(s) for figure 6:

**Source data 1.** Data and scripts for *Figure 6* and related figure supplements.

**Figure supplement 1.** Controls for promoter region mutants.

To examine if promoter regions other than the ASE motif were responsible for the differences in *che-1* and *gcy-22* expression, we replaced ASE motifs together with 59 bp flanks on either side (*Figure 6A*). (ASE*gcy-22*+ flanks)*p::che-1::GFP::AID* animals exhibited a strong chemotaxis defect and lack of CHE-1::GFP::AID expression in ASE neurons of larvae (*Figure 6B*, *Figure 6—figure supplement 1B*). At the (pre-)bean stage, (ASE*gcy-22*+ flanks)*p::che-1::GFP::AID* embryos showed high expression of *che-1::GFP::AID*, 28 ± 7 mRNA/cell, and CHE-1::GFP::AID (*Figure 6C and D*, *Figure 6—figure supplement 1B*), indicating that *che-1::GFP::AID* expression was initiated normally during ASE determination. This phenotype was also seen in the *che-1(p679)* loss of function mutant (*Figure 6—figure supplement 1C*). However, *che-1::GFP::AID* mRNA was absent in (ASE*gcy-22*+ flanks)*p::che-1::GFP::AID* embryos at later stages, indicating that this 130 bp *che-1* promoter region is not required for initiation of *che-1* expression but is important for maintenance of *che-1* expression and ASE fate. (ASE*che-1*+ flanks)*p::gcy-22* animals showed wild-type chemotaxis to NaCl (*Figure 6B*) and *gcy-22* mRNA levels (*Figure 6F*). Yet, strikingly, upon CHE-1::GFP::AID depletion in (ASE*che-1*+ flanks)*p::gcy-22* animals *gcy-22* mRNA levels remained high (*Figure 6F*), indicating that, like *che-1*, *gcy*-22 expression became resilient to CHE-1 depletion. Hence, the 130 bp *che-1* promoter fragment surrounding the ASE motif is responsible for maintaining expression during CHE-1 depletion.

## Involvement of an *Otx*-related homeodomain binding site in maintaining ASE fate

Within the 130 bp *che-1* promoter fragment required for resilient *che-1* expression, we identified a high scoring *Otx*-related homeodomain transcription factor (HD-TF) binding site, 29 bp downstream of the *che-1* ASE motif (*Figure 7A*). Only ~60 out of ~1000 CHE-1 targets exhibit a binding site of similar score within 100 bp of their ASE motifs (*Supplementary file 2*). To test the function of this HD-TF binding site, we deleted it from the *che-1* promoter in the *che-1::GFP::AID* background. These (ΔHD)*p::che-1::GFP::AID* animals showed an intermediate chemotaxis defect (*Figure 7B*). To examine whether this defect reflected changes in *che-1::GFP::AID* expression, we scored CHE-1::GFP::AID expression at different larval stages (*Figure 7C*). CHE-1::GFP::AID was expressed in all embryos, but was progressively lost over time, with CHE-1::GFP::AID absent in more than half of the young adults (2/18 animals for ASEL and 9/20 for ASER), while CHE-1::GFP::AID was always present in both ASE neurons of *che-1::GFP::AID* young adults (n = 23 animals). This indicated a defect in maintenance of *che-1* expression, not in ASE determination. We then used time-lapse microscopy (*Gritti et al., 2016*) to monitor the dynamics of CHE-1::GFP::AID expression directly in single (ΔHD)*p::che-1::GFP::AID* larvae (*Figure 7—figure supplement 1A*). Strikingly, CHE-1::GFP::AID expression was lost in a rapid manner at random times during larval development (*Figure 7E*, *Figure 7—figure supplement 1A*), as expected for spontaneous, noise-driven transitions to the OFF state. Theoretical studies showed that the rate of such transitions increases dramatically with decreasing copy number of the key transcription factors involved (*Warren and ten Wolde, 2005*). Indeed, (ΔHD)*p::che-1::GFP::AID* animals showed lower CHE-1::GFP::AID fluorescence, corresponding to 190 ± 70 CHE-1::GFP:AID proteins/cell (*Figure 7—figure supplement 1B*), and decreased *che-1::GFP::AID* mRNA levels, with 1 ± 1 mRNAs/cell in L1 larvae (*Figure 7D*), both considerably lower than seen in wild-type and *che-1::GFP::AID* animals (*Figures 2B and 5B*, *Figure 7—figure supplement 1B, E*). In addition, *che-1::GFP::AID* expression was lost more often and earlier in development in ASEL neurons (*Figure 7C and E*), which have lower average *che-1* mRNA and protein copy numbers than ASER neurons (*Figure 2D and F*). These results suggest that homeodomain proteins binding the HD-TF-binding site are essential for long-term maintenance of *che-1* expression and thus ASE cell fate, presumably by protecting the ON state against low CHE-1 copy number fluctuations.

The TF with highest predicted affinity to the HD-TF binding site in the *che-1* promoter (Materials and methods), and expressed in ASE neurons, is the *Otx*-related homeodomain transcription factor CEH-36. Previously, *ceh-36* mutations were shown to impact ASE specification and function (*Chang et al., 2003*; *Lanjuin et al., 2003*; *Walton et al., 2015*). To test whether CEH-36 is the key TF binding the HD-TF binding site, we generated a *ceh-36(gj2127)* deletion allele using CRISPR/Cas9, that deleted the full *ceh-36* coding region, in the *che-1::GFP::AID* background. This *ceh-36(gj2127)* deletion allele displayed a stronger salt chemotaxis defect than the *ceh-36(ks86)* missense allele (*Figure 7—figure supplement 1D*). However, when we examined *ceh-36(gj2127)* animals for the presence of CHE-1::GFP::AID, we found that, in contrast to (ΔHD)*p::che-1::GFP::AID* animals, CHE-1::GFP::AID was

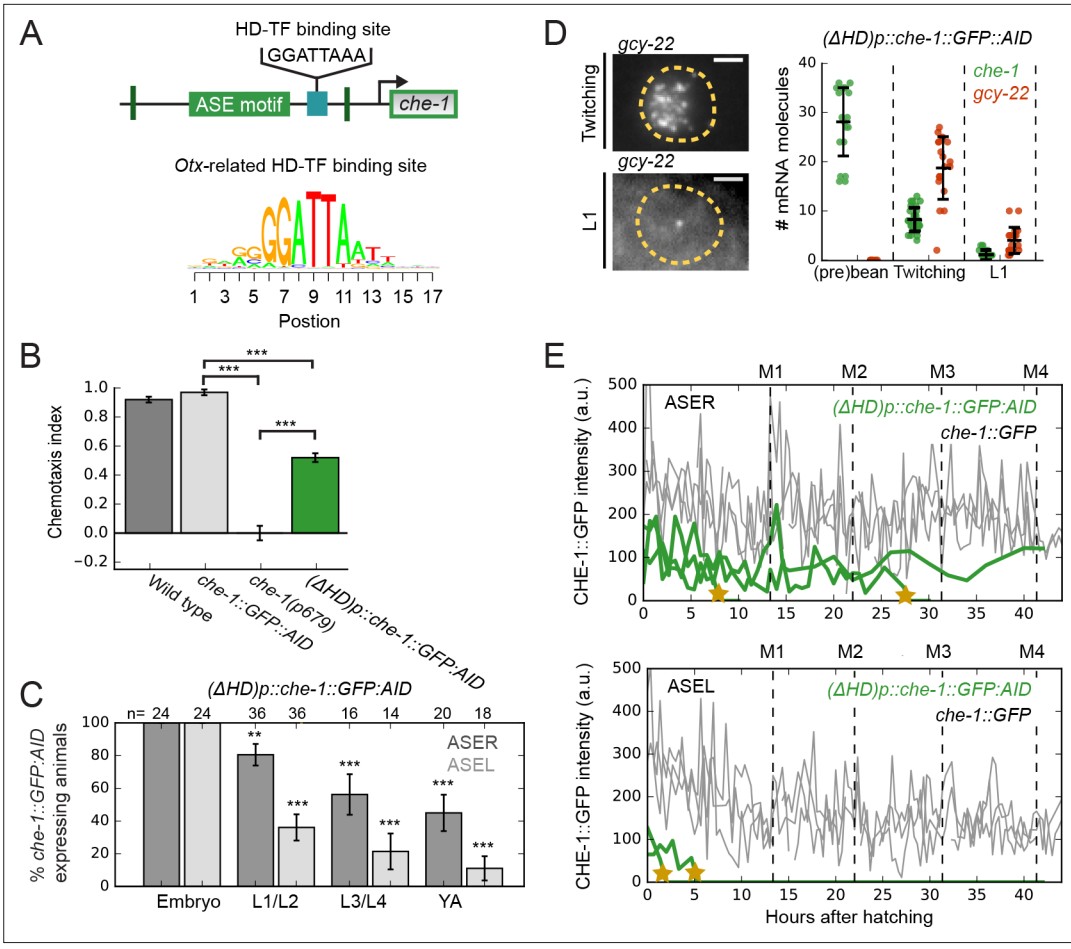

**Figure 7.** An *Otx*-related homeodomain transcription factor binding site involved in long-term maintenance of ASE cell fate. (**A**) Position of an *Otx*-related homeodomain transcription factor (HD-TF) binding site in the *che-1* promoter. Green lines indicate the positions of the 59 bp flanks surrounding the ASE motif. HD-TF binding site depicted as a sequence logo. In (*ΔHD)p::che-1::GFP::AID* animals, the HD-TF-binding site is deleted in the *che-1::GFP::AID* background. (**B**) Average chemotaxis index for response to 10 mM NaCl, of wild-type, *che-1::GFP::AID* and *che-1(p679)* animals, and (*ΔHD)p::che-1::GFP::AID* mutant. Deleting the HD-TF-binding site caused a decreased response to NaCl. (**C**) Fraction of (*ΔHD)p::che-1::GFP::AID* animals expressing CHE-1::GFP::AID in ASER (dark gray) and ASEL (light gray) at different developmental stages. CHE-1::GFP::AID is progressively lost during development. (**D**) *che-1::GFP::AID* and *gcy-22* mRNA levels in (*ΔHD)p::che-1::GFP::AID* animals quantified by smFISH. Expression was similar to wild-type until late-stage, twitching embryos, but fell rapidly in newly-hatched L1 larvae. Scale bar: 2 μm. (**E**) CHE-1::GFP expression dynamics in single *che-1::GFP* (grey) and (*ΔHD)p::che-1::GFP::AID* (green) animals during larval development. Approximate timing of molts is indicated M1-M4. CHE-1::GFP::AID expression in (*ΔHD)p::che-1::GFP::AID* animals was lost in a rapid and stochastic manner, at different times during development. Error bars represent S.E.M (**B,D**) or S.D. (**C**). n ≥ 10. **p < 0.01, ***p < 0.001.

The online version of this article includes the following source data and figure supplement(s) for figure 7:

**Source data 1.** Data and scripts for *Figure 7* and related figure supplements.

**Figure supplement 1.** Role of HD-TF-binding site in maintaining CHE-1::GFP expression.

**Figure supplement 2.** Models of HD-TF action.

present in all larvae, from L1 to adulthood (*Figure 7—figure supplement 1C*). This result might indicate that CEH-36 acts redundantly with other HD-TFs known to be expressed in ASE neurons (*Reilly et al., 2020*).

These results raise the question how HD-TFs impact *che-1* gene expression. An attractive model is that HD-TFs act as co-factors that increase the residence time of CHE-1 specifically at the *che-1* promoter, and thereby maintain *che-1* expression (*Figure 7—figure supplement 2A*). In this model,

we assumed that the unbinding rate of CHE-1 from the *che-1* promoter is decreased 1000-fold when an HD-TF is bound to the HD-TF binding site. For these parameters, the mass action rate equations show bistability, just as the original model without HD-TFs. When we performed stochastic simulations, this model could reproduce three key experiments: the maintenance of *che-1* expression following a transient inductive NHR-67 signal (*Sarin et al., 2009*), the failure to maintain *che-1* expression in ASE motif mutants where CHE-1 cannot bind to its own promoter (*Leyva-Díaz and Hobert, 2019*) and the resilience of *che-1* expression to transient CHE-1 depletion (*Figure 7—figure supplement 2B-D*). In contrast, an alternative model where HD-TFs act not as co-factors, but instead induce *che-1* expression independently of CHE-1, could reproduce *che-1* maintenance and resilience, but not the failure to maintain *che-1* in ASE motif mutants. These results therefore suggest that HD-TFs do not directly control *che-1* expression, but rather function to increase affinity of CHE-1 specifically for its own promoter.

## Discussion

The terminal selector gene *che-1* controls neuron type determination of the salt-sensing ASE neurons, by inducing the expression of hundreds of ASE-specific target genes while also inducing its own expression via autoregulation (*Etchberger et al., 2007*). A previous study showed that upon inhibition of positive CHE-1 autoregulation, transient *che-1* induction did not result in sustained *che-1* expression, with ASE cell fate lost shortly after its determination (*Leyva-Díaz and Hobert, 2019*). This result left open whether this positive feedback loop by itself is sufficient to maintain *che-1* expression and ASE type for the animal's entire lifetime, or whether additional mechanisms, such as chromatin modifications, are required after initial determination to lock in cell identity in an irreversible manner. Here, we show that *transient* depletion of CHE-1 is sufficient to permanently lose ASE function, indicating that indeed *che-1* expression forms a reversible, bistable switch. This also raised the question how, in the presence of the inherent molecular fluctuations in the cell, this switch maintains its ON state (*che-1* expression) and prevents spontaneous transitions to the OFF state (no *che-1* expression).

Theoretical studies found that the stability of bistable genetic switches against fluctuations is enhanced by increasing the copy number and lifetime of the transcription factors that form the switch (*Mehta et al., 2008*; *Walczak et al., 2005*; *Warren and ten Wolde, 2005*). However, our simulations using in vivo measured *che-1* mRNA and protein copy numbers and lifetimes showed that these values by themselves were not sufficient to generate a stable ON state. Instead, our simulations suggested a novel mechanism required to stabilize the ON state, which uses the reservoir of CHE-1 proteins bound to its many target sites to buffer *che-1* expression against fluctuations in CHE-1 level. Crucial to this target reservoir buffering mechanism is that CHE-1 exhibits strong preferential binding to its own promoter compared to its other targets. In this case, when CHE-1 protein levels drop, any CHE-1 protein that dissociates from a target gene promoter will immediately bind the *che-1* promoter, if it is unoccupied. This ensures maintenance of *che-1* expression, at the expense of expression of its other targets, to bring CHE-1 protein levels back to the normal steady state. Our experimental observations verified a key prediction following from this mechanism: upon transient CHE-1 depletion in vivo, expression of the CHE-1 target *gcy-22* rapidly and strongly decreased, yet *che-1* expression itself was hardly affected. We show that this resilience of *che-1* expression to CHE-1 depletion could be conferred onto other target genes by introducing a 130 bp fragment of the *che-1* promoter that surrounds the ASE motif bound by CHE-1, but not the ASE motif itself.

A definitive test of the target reservoir buffering mechanism would require direct measurements of CHE-1 binding to the promoter of *che-1* and its other targets. Transcription factor binding affinities can be measured in vitro (*Etchberger et al., 2007*), but such assays likely miss essential co-factors or chromatin states that control CHE-1 binding in vivo. Other techniques, such as ChIP-seq, can measure transcription factor binding in vivo in a genome-wide manner (*Askjaer et al., 2014*). However, while ChIP-seq analysis was used recently to characterize genome-wide binding of EGL-43, a transcription factor expressed in 36 neurons and the somatic gonad, the comparatively lower protein levels of CHE-1, present in only two ASE neurons per animal, make it virtually impossible to interpret such ChIP-seq data quantitatively in terms of binding affinities (*Deng et al., 2020*). Nevertheless, we expect that techniques such as ChIP-seq or CUT&RUN (*Skene and Henikoff, 2017*) could be used in the future to validate key predictions of the target reservoir mechanism, particularly in combination with specifically isolating ASE neurons (*Kaletsky et al., 2016*).

Target reservoir buffering relies on CHE-1 being preferentially recruited to the *che-1* promoter. If this mechanism is indeed responsible for stable ASE fate maintenance, our work suggests that homeodomain transcription factors are involved, as deleting a HD-TF binding site close to the *che-1* ASE motif and specific to the *che-1* promoter, caused spontaneous transitions to the OFF state, and loss of cell identity, long after ASE type determination. This lack of stability was accompanied by lower *che-1* mRNA and protein levels, consistent with reduced recruitment of *che-1* to its own promoter. Our simulations show that HD-TFs might act as co-factors that increase the residence time of CHE-1 at the *che-1* promoter. Our theoretical estimates of ON state lifetimes indicate that a highly stable ON state requires a relatively small, 10–100 fold increase in CHE-1 residence time at the *che-1* promoter compared to its other targets, within the range expected if CHE-1 and a HD-TF interact cooperatively with a weak interaction of a few $k_B$T. A similar interaction was postulated for the homeodomain protein ALR-1 in mechanosensory TRN neurons, where it restricts variability in expression of the terminal selector gene *mec-3*, by binding close to the MEC-3 binding site on its promoter (*Topalidou et al., 2011*). Five HD-TF are expressed in both ASE neurons, *ceh-36*, *ceh-54*, *ceh-79*, *ceh-89*, and *dsc-1*, and four more are expressed only in ASER, *cog-1*, or ASEL, *lim-6*, *alr-1*, and *ceh-23* (*Reilly et al., 2020*). Even though CEH-36 has high predicted affinity for the HD-TF binding site and *ceh-36* mutants display chemotaxis defects indicative of a role in ASE function, a *ceh-36(gj2127)* deletion mutant did not reproduce the *che-1* expression defect seen upon deletion of the HD-TF binding site, potentially indicating a role for other ASE-expressed homeodomain proteins in combination with CEH-36. Further analyses are required to test whether these HD-TFs or a combination of them indeed play a role in target reservoir buffering.

The observed resilience of *che-1* expression to CHE-1 depletion is crucial for stable maintenance of ASE identity. In both our models with and without cooperativity, this is achieved by a very low threshold for inducing *che-1* expression, with a single *che-1* mRNA potentially sufficient to induce the ON state. Indeed, the recovery of CHE-1 expression after 24–48 hr of CHE-1 depletion demonstrates that the ON state can be recovered from very low CHE-1 levels. However, this raises the question how stochastic, spontaneous induction of *che-1* expression and ASE fate is prevented in non-ASE cells. In most cells, spontaneous induction of the ON state is likely prevented by chromatin remodelling (*Patel and Hobert, 2017*; *Tursun et al., 2011*). However, transient *che-1* expression induced long-term expression of CHE-1 targets in non-ASE head sensory neurons, suggesting that these cells are capable of inducing the ON state (*Tursun et al., 2011*). A potential mechanism to prevent ectopic induction of the ON state is provided by our observation that preferential binding of CHE-1 to its own promoter likely depends on homeodomain proteins. Cells lacking these proteins would have difficulty inducing and maintaining *che-1* expression. Consistent with this hypothesis, a large number of HD-TFs is expressed in the *C. elegans* nervous system in a highly neuron-specific manner (*Hobert, 2010*; *Reilly et al., 2020*). We hypothesize that cell-specific expression of co-factors of terminal selector genes might form a general mechanism to prevent spontaneous induction of these genes in the wrong cells.

While bistability in genetic networks is recognized as an important mechanism to generate cell fate switches (*Ferrell, 2002*), long-term cell fate maintenance is often assumed to require additional feedback mechanisms, for instance through histone and chromatin modifications, that make cell fate essentially irreversible. Here, we show that bistability through an autoregulatory feedback loop alone is sufficient for life-long maintenance of neuron identity in *C. elegans*, despite strong stochastic molecular fluctuations in the underlying genetic network. The mechanism we propose for achieving this, target reservoir buffering, depends crucially on the single-input module topology of the network, with a terminal selector, CHE-1, inducing both its own expression and that of many other target genes. Single-input modules are found in network motifs for neuron type determination in both *C. elegans* and higher organisms (*Hobert and Kratsios, 2019*). In addition, cell differentiation in general is often controlled by a small number of master regulators that, directly or indirectly, induce both their own expression and that of many cell fate-specific target genes. Hence, we expect target reservoir buffering to play an important, general role in explaining stable long-term maintenance of cell fate in a broad array of systems.

## Materials and methods
### *C. elegans* strains and handling
The following alleles were used in this study: *ceh-36(gj2127) X*, *ceh-36(ks86) X*, *che-1(p679) I*, *che-1(ot856[che-1::GFP]) I* (kindly provided by Dylan Rahe from the Hobert lab) (*Leyva-Díaz and*

*Hobert, 2019*), *che-1(gj2089[che-1::GFP::AID])* I, *che-1(gj2088[(ΔHD)p::che-1::GFP::AID])* I, *che-1(gj2063[(ASE_gcy-22)p::che-1::GFP::AID])* I, *che-1(gj2062[(ASE_gcy-22+ flanks)p::che-1::GFP::AID])* I, *gcy-5(ot835[gcy-5::SL2::mNeonGreen])* II, *osm-3(gj1959[osm-3::GFP])IV*, *gcy-22(gj2064[(ASE_che-1)p::gcy-22])* V, *gcy-22(gj2065[(ASE_che-1+ flanks)p::gcy-22])* V, *ieSi57[eft-3p::TIR1::mRuby::unc-54 3'UTR, Cbr-unc-119(+)] II* (*Zhang et al., 2015*), *ntIs1 [gcy-5p::GFP+ lin-15(+)] V, otIs3 [gcy-7::GFP+ lin-15(+)] V, otTi6[hsp16-41p::che-1::2xFLAG] X*. The wild-type strain used was the *C. elegans* variety Bristol, strain N2. All *C. elegans* strains were maintained on Nematode Growth Medium (NGM) plates, containing *E. coli* strain OP50 as a food source, at 20 °C (*Brenner, 1974*), unless indicated otherwise. Worms were maintained according to standard protocol.

## Molecular biology

To generate the *pU6::osm-3_sgRNA* vector we cloned an *osm-3* guide into the *pU6::unc-119::sgRNA* vector (*Friedland et al., 2013*; *van der Burght et al., 2020*). The *osm-3::GFP* template construct was generated by inserting GFP, amplified from pPD95.77 (gift from A. Fire) and two 1.5 kb homology arms, amplified from genomic DNA using primers #2,679 and #2,643 and primers #2,660 and #2646, into the backbone of *pU6::unc-119::sgRNA*.

## CRISPR/Cas9-mediated genome editing

Genome editing was performed according to protocol (*Dokshin et al., 2018*) and using ssODN repair templates with 35 bp homology arms (*Paix et al., 2017*). An AID tag was endogenously inserted at the C-terminus of GFP in a *che-1(ot856[che-1::GFP])* background using guide g2 and a repair template containing the degron sequence (*Zhang et al., 2015*), generating *che-1(gj2089[che-1::GFP::AID])*. Subsequently, the *ieSi57* allele (*Zhang et al., 2015*) was introduced. All *che-1* or *gcy-22* promoter mutations were made in this *che-1(gj2089[che-1::GFP::AID]; ieSi57* background. (*ASE_gcy-22+ flanks)p::che-1::GFP::AID* was generated using a template containing the ASE motif of the *gcy-22* promoter and its 59 bp flanking regions, and guides g51 and g52. The choice of 2 × 59 bp flanking regions was motivated by the maximum oligo length of 200 bp, taking into account the length of the homology arms (35 bp) and the ASE motif itself (12 bp). (*ASE_che-1+ flanks)p::gcy-22* was generated using a template containing the ASE motif of *che-1* and its 59 bp flanks and guides g55 and g56. (*ASE_gcy-22)p::che-1::GFP::AID* was generated using a template containing the ASE motif of *gcy-22* and guide g53. (*ASE_che-1)p::gcy-22* was generated using a template containing the ASE motif of *che-1* and guide g57. (*ΔHD)p::che-1::GFP::AID* was generated using a template containing the 35 bp flanking regions of the HD motif from the *che-1* promoter and guide g54. The *ceh-36(gj2127)* deletion allele was generated using repair template 3,579 and guides g63 and g66. To generate the *osm-3::GFP* allele, animals were injected with a mixture containing *p_U6::osm-3_sgRNA* (50 ng/ml), *p_eft-3::cas9-SV40_NLS::tbb-2* (50 ng/ml), pRF4::*rol-6(su1006)* (50 ng/ml), and the *osm-3::GFP* repair template (20 ng/ml). Animals were injected and placed on separate 6 cm NGM plates. Three days later, F1 offspring was picked, allowed to self-reproduce, and screened by PCR. Guides and ssODNs used in this research are listed in *Supplementary file 4*.

## Single molecule fluorescence in situ hybridization (smFISH)

The oligonucleotides for the smFISH probe sets were designed with optimal GC content and specificity for the gene of interest using the Stellaris RNA FISH probe designer. The oligonucleotides were synthesized with a 3′ amino C7 modification and purified by LGC Biosearch Technologies. Conjugation of the oligonucleotides with either Cy5 (GE Amersham) or Alexa594 (Invitrogen) was done as previously described (a). Sequences of each probe set are listed in *Supplementary file 3* (with exception of *gcy-14*). The smFISH protocol was performed as previously described (*Ji and van Oudenaarden, 2012*; *Raj et al., 2008*). Briefly, staged animals were washed from plates with M9 buffer and fixed in 4% formaldehyde in 1 x PBS, gently rocking at room temperature (RT) for 40 min (young adults for 35 min). Fixation of embryos required a snap-freeze step to crack the eggshells by submerging embryos, after 15 min in fixation buffer, in liquid nitrogen, and thawing on ice for 20 min. After fixation, the animals were 2 x washed with 1xPBS and resuspended in 70% ethanol overnight at 4 °C. Ethanol was removed and animals were washed with 10% formamide and 2 X SSC, as preparation for the hybridization. Animals were incubated with the smFISH probes overnight in the dark at 37 °C in a hybridization solution (Stellaris) with added 10% formamide. The next day, animals were washed 2 x with 10% formamide and 2 X SSC each with an incubation step of 30 min at 37 °C. The

last wash step contains DAPI 5 μg/mL for nuclear staining. The wash buffer was removed, and animals were resuspended in 2 X SSC and stored at 4 °C until imaging. The 2 X SSC was aspirated and animals were immersed in 100 μl GLOX buffer (0.4% glucose, 10 mM Tris-HCl, pH 8.0, 2 X SSC) together with 1 μl Catalase (Sigma-Aldrich) and 1 μl glucose oxidase (Sigma-Aldrich) (3.7 mg/mL) to prevent bleaching during imaging.

Microscopy images of smFISH samples were acquired with a Nikon Ti-E inverted fluorescence microscope, equipped with a 100 X plan-apochromat oil-immersion objective and an Andor Ikon-M CCD camera controlled by μManager software (*Edelstein et al., 2014*). smFISH analysis was performed with custom Python software, based on a previously described method (*Raj et al., 2008*). Briefly, we first convolved the smFISH images with a Gaussian filter. Next, candidate spots were selected via manual thresholding, and partially overlapping spots were separated via finding 3D regional intensity maxima. We used the spatial localization of *gcy-22* or *gcy-14* mRNA molecules which are highly expressed ASE-specific genes, to estimate the cell boundaries of the ASE neurons. The coverage of the ASE cell bodies with *gcy-22* or *gcy-14* mRNA molecules agreed with GFP markers (*gcy-5p::GFP* or *gcy-7p::GFP*) that marked the cell body.

## CHE-1 protein quantification

Both *che-1::GFP* and *che-1::GFP::AID* animals were homozygous for the tag-insertions, so that GFP fluorescence visualized the entire pool of expressed CHE-1::GFP or CHE-1::GFP::AID. We therefore assumed that GFP fluorescence scaled linearly with CHE-1 abundance. To calibrate GFP fluorescence in terms of protein copy number, staged *che-1::GFP* knock-in animals were bathed in 72 nM and 48 nM eGFP recombinant protein (Bio-connect) with 0.25 mM Levamisole (Sigma-Aldrich) in M9 buffer in a glass chambered cover glass systems (IBL baustoff), which were coated with 0.5 mg/ml kappa-capsein in the buffer MRB80 (80 mM Pipes, 4 mM MgCl2, 1 mM EGTA, pH 6.8 with KOH) to prevent binding of eGFP to the cover glass and chamber walls. Images of bathed *che-1::GFP* animals in eGFP solution were acquired with a Nikon Eclipse Ti inverted microscope, equipped with a Nikon C1 confocal scan head, a 100 mW Argon ion laser (488 nm, Coherent), and a S Fluor 40 × 1.3 NA and an Apo TIRF 100 × 1.49 NA objective. Calibration of eGFP with *che-1::GFP* animals was repeated two times at different days with 72 nM and 48 nM eGFP concentrations, of which we took the average calibration measurements. For ease of measuring, the CHE-1::GFP signal of animals was measured with the exact same microscope and software settings, except placing the animals submersed in 0.25 mM Levamisole (Sigma-Aldrich) in M9 buffer on agar pads with the same cover glass thickness on top. The ASE neuron closest to the cover glass was imaged in larvae to circumvent tissue scattering. Embryos were followed in time (at 22°C) and imaged every 20 min with the exact same microscope and software settings, from bean stage until twitching started. For both larvae and embryos, the slices focused at the approximate middle of the ASE neuron nuclei were used for quantifying the CHE-1::GFP signal. The volumes of the nuclei were calculated by measuring the radii of the nuclei in x, y and z direction from the CHE-1::GFP signal with the assumption that the nucleus shape resembles a ellipsoid, using the following equation: $V = \frac{4}{3}\pi xyz$.

## FRAP

To estimate the protein degradation rate of CHE-1::GFP, we used Fluorescence Recovery After Photobleaching (FRAP). Animals were immobilized using Polybead microspheres (Polyscience) in M9 buffer on agarose pads covered with a cover glass. To prevent dehydration of animals, the coverslip was sealed with VALAP (vaseline, lanolin and paraffin, ratio 1:1:1). Animals were monitored at several time points during the experiment if they were still alive by checking very subtle movement and/or pumping behaviour. The GFP signal in the ASE neurons of the animals was bleached until approximately 20% of the initial signal was left. After bleaching, the GFP signal was measured every 20 or 30 min until the signal had recovered. Images were taken with the same microscope as in the CHE-1 protein quantification section. We measured for each time point the average GFP intensity in the ASE neurons and subtracted the background intensity measured nearby the ASE neurons. The degradation rate was calculated from the initial slope of the growth curve using the following exponential model: $R(t) = \left(\frac{f}{b}\right) + \left(x_0 - \left(\frac{f}{b}\right)\right) e^{-bt}$, where $x_0$ is the initial fluorescent intensity at the start of the recovery curve right after bleaching. $b$ and $f$ represent the CHE-1 protein degradation and production rate, which are fitted on the individual measured recovery curves, to obtain the average CHE-1 protein degradation rate.

### *che-1* mRNA stability

We induced *che-1* mRNA overexpression with the *otTi6 [hsp16-41p::che-1::2xFLAG] X* inducible heat shock strain, and we quantified with smFISH *che-1* mRNA in the ASER neurons in L2 larvae over time ($t_i = 0, \ldots, 4$) ~ 17 min apart until recovery. We determined the relative amount of *che-1* mRNAs from the start of the measurement, $n\left(t_i\right) = \frac{N\left(t_i\right) - N_{control}}{N_{HS} - N_{control}}$, where $N_{control}$ is the calculated average amount of *che-1* mRNAs when there is no *che-1* mRNA overexpression, $N_{HS}$ the calculated average amount of *che-1* mRNAs at the first measured time point right after heat shock induction, and $n\left(t\right)$ is the amount of *che-1* mRNAs at the three remaining time points. An exponential degradation curve $e^{-at}$ was fitted to the experimentally determined values $n\left(t_i\right)$, to obtain the approximate *che-1* mRNA degradation rate.

## Mathematical model of the CHE-1 switch

### Overview cooperative model

The cooperative mathematical model assumes that CHE-1 protein ($C$) has to bind as a dimer on the *che-1* promoter to induce *che-1* mRNA expression. The binding and unbinding of the two CHE-1 proteins at the *che-1* promoter ($O$) is separated into two events, first one CHE-1 binds with binding rate $f_O$ and unbinds with unbinding rate $b_1$ from the *che-1* promoter ($OC$). When the first CHE-1 protein is bound to the *che-1* promoter, the second CHE-1 protein binds with the same binding rate $f_O$ next to the first CHE-1, forming a dimer on the *che-1* promoter ($OC2$), and unbind3s with the unbinding rate $b_2$. CHE-1 proteins bind as a monomer on the target gene promoters ($O_TC$), with the unbinding rate $b_T$, and with same the binding rate $f_O$ as on the *che-1* promoter. *che-1* mRNA ($M$) is transcribed with the production rate $f_M$ only when two CHE-1 proteins are bound to the *che-1* promoter, and *che-1* mRNA is translated into CHE-1 protein with the protein production rate $f_C$. Both *che-1* mRNA and CHE-1 protein are degraded with the degradation rates, $b_m$ and $b_C$ respectivily. This leads to the following differential equations:

$$\frac{dOC}{dt} = f_O\left(O^* - \left(OC + OC2\right)\right)C - b_1 OC$$

$$\frac{dOC2}{dt} = f_O OC \cdot C - b_2 OC2$$

$$\frac{dO_TC}{dt} = f_O\left(O_T^* - O_TC\right)C - b_T O_T \cdot C$$

$$\frac{dM}{dt} = f_M OC2 - b_M M$$

$$\frac{dC}{dt} = -f_O\left(O^* - \left(OC + OC2\right)\right)C - f_O OC \cdot C - f_O\left(O_T^* - O_TC\right)C + b_1 OC + b_2 OC2 + b_T O_T \cdot C + f_c M - b_C C$$

where $O^*$ is the total number of *che-1* promoters and where $O_T^*$ is the total number of target genes.

### Overview non-cooperative model

In the non-cooperative model, CHE-1 binds ($f_O$) and unbinds ($b_O$) as a monomer on the *che-1* promoter ($OC$) and induces *che-1* expression as a monomer. The other reactions in the model are the same as in the cooperative model. This leads to the following differential equations:

$$\frac{dOC}{dt} = f_O\left(O^* - OC\right)C - b_O OC$$

$$\frac{dO_TC}{dt} = f_O\left(O_T^* - O_TC\right)C - b_T O_T \cdot C$$

$$\frac{dM}{dt} = f_M OC - b_M M$$

$$\frac{dC}{dt} = -f_O\left(O^* - OC\right)C - f_O\left(O_T^* - O_TC\right)C + b_O OC + b_T O_T.C + f_c M - b_C C$$

### Bistability

In the cooperative model, CHE-1 proteins bind as a dimer at the *che-1* promoter to induce *che-1* expression. The binding of two CHE-1 proteins in the system, implies non-linear behaviour, giving rise to bistability. We have in the cooperative model three fixed points, two stable and one unstable fixed point. The two stable fixed points represent the so-called 'ON state' (high CHE-1) and the 'OFF state' (low CHE-1) of the CHE-1 switch. When the switch is in the OFF state, it has to cross the unstable point to reach the ON state. In contrast, the non-cooperative model, in which CHE-1 proteins bind as a monomer at the *che-1* promoter to induce *che-1* expression, has only two fixed points. The first fixed point represents the ON state, in which CHE-1 protein levels are high. The second fixed point is a half-stable point, when there is no CHE-1 protein,

the switch is OFF. However, the introduction of, for example, only one *che-1* mRNA in the system would be sufficient to turn the switch from the OFF state to the ON state.

## Simulations

To study the short-term lifetime ( < 1 weeks) of the cooperative and non-cooperative CHE-1 switch, we performed stochastic Gillespie simulations on both models (*Gillespie, 2002*). We used a custom written python script to simulate the reactions involved in both CHE-1 switch models. All reactions describing the differential equations from the cooperative and non-cooperative model, are summarized in *Figure 3A*. Parameters remain unchanged during simulations (with exception for transient depletion simulations) and species are initiated in the ON state.

To study ON state lifetimes of the CHE-1 switch exceeding 1 weeks, in the cooperative and non-cooperative CHE-1 switch model, we used Forward Flux Sampling (*Allen et al., 2006*; *Allen et al., 2009*), a computational method which allowed us to estimate lifetimes of CHE-1 switches. It was not necessary to integrate pruning into the algorithm since this would not result in improvement in computational efficiency. The FFS algorithm was initiated with the same initial conditions as the Gillespie simulations. Interfaces of the FFS algorithm were chosen according to the variance of CHE-1 protein in the ON state, to generate a 5–10% chance of CHE-1 protein trajectories crossing the first interface (with an exception for very unstable switch, where CHE-1 protein simulations immediately run to the OFF state). The typical step size of the interfaces was 20 and the number of interfaces was between 20 and 35.

## Parameters

We can divide the parameters in the following groups: (*1*) *experimentally determined parameters*, (*2*) *parameters that we could approximate*, (*3*) *unconstrained parameters,* and (*4*) *parameters that we could calculate with help of the other parameters*. First, the parameters which were experimentally determined. We based the *che-1* mRNA degradation rate ( $b_M$) on direct measurement of *che-1* mRNA degradation that we quantified in the ASER neuron (*Figure 2A–B*). We used an approximation of *che-1* mRNA lifetime of 20 min. The amount of CHE-1 protein $(C)$ was set to 900 molecules (average of ASER and ASEL at L4/YA stage) based on the CHE-1 protein quantification experiments, and the amount of *che-1* mRNAs $(M)$ was set to seven molecules (average of ASER and ASEL at L4 stage) based on the smFISH experiments in wild-type animals.

The binding rate of CHE-1 at its own promoter $f_O$ and the target gene promoters $f_O$ is approximated by the diffusion limited binding rate. The diffusion limited binding rate of CHE-1 on promoter sites was calculated using the following diffusion equation: $f_D = 4\pi\sigma D$, where reaction cross section $\sigma = 1 \cdot 10^{-2} \ \mu m$, that is, the size of the promoter binding site, and the diffusion coefficient constant is $D = 1 \ \mu m^2 \ s^{-1}$ . To obtain the diffusion coefficient that we can apply in our simulations, which includes information about the volume of the nucleus, the diffusion coefficient $(f_D)$ was divided with the average nucleus volume $(V_C)$ of $4 \ \mu m^3$ in ASER and ASEL, resulting in a diffusion limited binding rate of $f_O = \frac{f_D}{V_C} = 0.03 \ s^{-1}$ .

To approximate the number of target sites where the CHE-1 protein can bind, we used the crude approximation depicted from the study of *Etchberger et al., 2007*, in which they found 596 genes represented with at least three times as many tags in the ASE versus AFD SAGE library, which is expected to contain almost no 'false positives'. We acquired the total amount of ASE motifs (minimum score of 98%) in each of the promoter regions ( < 1000 bp upstream of the start site) with a custom written R script, using the TFBSTools/JASPAR2018 packages, resulting in a total of 425 ASE motifs. However, the data set lacked, for example, housekeeping genes expressed in the ASE neurons that could be under control of *che-1*. The sci-RNA-seq dataset from the study of Cao et al. provides information on genes expected to be expressed in the ASE neurons, including ASE non-specific genes (*Cao et al., 2017*). A total of 1400–1500 genes are expected to be expressed in the ASEL and ASER (score >100, removal of 'false-positives'), resulting in ~1000 ASE motifs on which CHE-1 could potentially bind. In the simulations we used either 500 or 1000 target sites.

The unbinding rates of CHE-1 from its own promoter and the target gene promoters, $b_1, b_2, \ b_T$ , are the only unconstrained parameters, and are ranged between 0.1 and 100 s⁻¹ in the FFS simulations. The CHE-1 protein degradation rate $b_C$ , the CHE-1 production rate $f_C$ and the *che-1* mRNA production rate $f_M$ are dependent on the unbound fraction of CHE-1 protein. To calculate the three unknown rates $b_C, f_C, f_M$ , first the CHE-1 protein degradation rate $b_C$ was set at the experimentally measured parameter, and used to

calculate the *che-1* mRNA production rate $f_M$ and CHE-1 protein production rate $f_C$. The CHE-1 production rate is influenced by the amount of CHE-1 bound at the target gene promoters, hence why the CHE-1 protein production and degradation had to be fitted to the experimentally measured CHE-1 protein FRAP curve. The fitting was done by introducing CHE-1::GFP species, bleached or unbleached, in the existing model in order to reproduce in simulation the FRAP experiments. The bleached CHE-1::GFP proteins could also bind and unbind the promoters and induce *che-1* expression, but the bleached CHE-1::GFP proteins could not be produced, only degraded.

In the table below, we summarize all parameter values. In this table, all parameter values are defined for the *Unstable CHE-1 switch (4B)*, and the other model parameter values are only given when they deviate from those used in the *Unstable CHE-1 switch (4B)*.

| Parameter | Description | Value |
|---|---|---|
| Unstable switch (4B) | Unstable CHE-1 switch, average lifetime of ~10 days | |
| $C$ | Number of CHE-1 protein molecules | 900 |
| $M$ | Number of *che-1* mRNA molecules | 7 |
| $O^*$ | Number of *che-1* promoters | 1 |
| $O_T^*$ | Number of target genes | 500 |
| $f_O$ | Binding rate of CHE-1 on *che-1* or target gene promoters | 0.03 s$^{-1}$ |
| $b_O$ | Unbinding rate of CHE-1 from *che-1* promoter | 100 s$^{-1}$ |
| $b_T$ | Unbinding rate of CHE-1 from target gene promoters | 100 s$^{-1}$ |
| $f_M$ | *che-1* mRNA production rate | 0.0302 s$^{-1}$ |
| $b_M$ | *che-1* mRNA degradation rate | 0.00083 s$^{-1}$ |
| $f_C$ | CHE-1 protein production rate | 0.0274 s$^{-1}$ |
| $b_C$ | CHE-1 protein degradation rate | 0.00024 s$^{-1}$ |
| Stable switch (4B) | Highly stable switch, slower unbinding rate of CHE-1 from its own promoter | |
| $b_O$ | Unbinding rate of CHE-1 from *che-1* promoter | 0.1 s$^{-1}$ |
| $f_M$ | *che-1* mRNA production rate | 0.0059 s$^{-1}$ |
| $f_C$ | CHE-1 protein production rate | 0.0261 s$^{-1}$ |
| $b_C$ | CHE-1 protein degradation rate | 0.00023 s$^{-1}$ |
| Stable switch (4E) | Highly stable switch, depleted to 100 CHE-1 | |
| $b_O$ | Unbinding rate of CHE-1 from *che-1* promoter | 0.1 s$^{-1}$ |
| $f_M$ | *che-1* mRNA production rate | 0.0059 s$^{-1}$ |
| $b_C$ | CHE-1 protein degradation rate | 0.0019 s$^{-1}$ |
| $f_C$ | CHE-1 protein production rate | 0.024 s$^{-1}$ |
| $b_P$ | Target mRNA degradation rate | 0.0008 s$^{-1}$ |
| $f_P$ | Target mRNA production rate | 0.09 s$^{-1}$ |
| Unstable switch (4E) | Unstable switch, depleted to 100 CHE-1 | |
| $b_O$ | Unbinding rate of CHE-1 from *che-1* promoter | 10 s$^{-1}$ |
| $b_T$ | Unbinding rate of CHE-1 from target gene promoters | 10 s$^{-1}$ |
| $b_C$ | CHE-1 protein degradation rate | 0.00076 s$^{-1}$ |
| $f_C$ | CHE-1 protein production rate | 0.026 s$^{-1}$ |
| $b_P$ | Target mRNA degradation rate | 0.0004 s$^{-1}$ |

*Continued on next page*

*Continued*

| Parameter | Description | Value |
|---|---|---|
| $f_P$ | Target mRNA production rate | $0.016 \text{ s}^{-1}$ |

The parameters for the depletion simulations panel (*Figure 4E*) include an extra promoter ($PO$) with production and degradation rates of ($PM$) mRNA, $b_P$, $f_P$, to simulate the production of target mRNA induced by CHE-1. The binding and unbinding rates of the extra promoter are the same as for the target genes, $f_O$, $b_T$.

## Overview CHE-1 model with HD-TF

In *Figure 7—figure supplement 2*, we examine a stochastic model, based on the non-cooperative model discussed above, that explicitly includes interactions with homeodomain transcription factors (HD-TFs), based on our identification of a HD-TF binding site required for *che-1* maintenance (*Figure 7*). We examine two different models: HD-TF acts as a co-factor that only impacts the affinity of CHE-1 for the *che-1* promoter (Model 1), or HD-TF induces *che-1* expression independent of CHE-1 (Model 2). For both models, we compared two variants: one where HD-TF expression is constitutive (Models 1 A, 2 A) and one where HD-TF expression is controlled by CHE-1 binding (Models 1B, 2B). The following reactions are common to both Models 1 and 2:

$$O + C \underset{f_{O,C}}{\overset{b_O}{\leftrightarrows}} OC \qquad O_T + C \underset{f_O}{\overset{b_O}{\leftrightarrows}} O_T C \qquad O + H \underset{f_{O,H}}{\overset{b_O}{\leftrightarrows}} OH$$

$$OC \underset{f_M}{\rightarrow} OC + M \qquad M \underset{f_C}{\rightarrow} M + C$$

$$M \underset{b_M}{\rightarrow} \varnothing \qquad C \underset{b_C}{\rightarrow} \varnothing$$

For Model 1, we have the following additional reactions:

$$OH + C \underset{f_{O,C}}{\overset{b_{O,s}}{\leftrightarrows}} OHC \underset{b_{O,s}}{\overset{f_{O,H}}{\leftrightarrows}} OC + H$$

$$OHC \underset{f_M}{\rightarrow} OHC + M$$

Here, the dissociation rate $b_{O,s} < b_O$ describes the proposed cooperative interaction between CHE-1 and HD-TFs on the *che-1* promoter, leading to higher affinity of CHE-1 to its own promoter if HD-TF is bound. For Model 2, instead we add the following reaction, that describes induction of che-1 expression by HD-TF independent of CHE-1:

$$OH \underset{f_{M,H}}{\rightarrow} OH + M$$

For Models 1B and 2B, we add the following reactions, that describe induction of HD-TF expression by CHE-1:

$$C \underset{f_H}{\rightarrow} C + H \qquad H \underset{b_H}{\rightarrow} \varnothing$$

For the simulation in *Figure 7—figure supplement 2D*, we add a reaction, $\varnothing \rightarrow M$, with rate $f_{M,0}$, to describe CHE-1-independent *che-1* expression induced during ASE specification. For parameters, we use copy numbers and rates corresponding to the *Unstable CHE-1 switch (4B)* defined above, reflecting that in absence of HD-TF the ON state is maintained with low stability.

*Continued on next page*

| Parameter | Description | Value |
|---|---|---|
| **Common** | | |
| $f_{O,C}$ | Binding rate of CHE-1 on *che-1* promoter | 0.03 s⁻¹ |
| $f_O$ | Binding rate of CHE-1 on target gene promoters | 0.03 s⁻¹ |
| $f_{O,H}$ | Binding rate of HD-TF on *che-1* promoter | 0.03 s⁻¹ |
| $b_O$ | Unbinding rate of CHE-1 and HD-TF from promoter | 100 s⁻¹ |
| $f_M$ | *che-1* mRNA production rate | 0.0085 s⁻¹ |
| $b_M$ | *che-1* mRNA degradation rate | 0.00083 s⁻¹ |
| $f_C$ | CHE-1 protein production rate | 0.0198 s⁻¹ |
| $b_C$ | CHE-1 protein degradation rate | 0.00032 s⁻¹ |
| $f_{M,0}$ | CHE-1 independent *che-1* mRNA production rate | 0.005 s⁻¹ |
| **Model 1** | | |
| $b_{O,s}$ | Slow unbinding rate of CHE-1/HD-TF from complex on promoter | 0.1 s⁻¹ |
| **Model 2** | | |
| $f_M$ | *che-1* mRNA production rate upon induction by CHE-1 | 0.01 s⁻¹ |
| $f_{M,H}$ | *che-1* mRNA production rate upon induction by HD-TF | 0.015 s⁻¹ |
| **Models 1B, 2B** | | |
| $f_H$ | HD-TF protein production rate | 0.0001 s⁻¹ |
| $b_H$ | HD-TF protein degradation rate | 0.000015 s⁻¹ |

To mimic the transient depletion experiment (*Figure 7—figure supplement 2B*), the degradation rate $b_C$ was increased 50-fold during the time of induced depletion. For all conditions apart from transient induction (gray areas in *Figure 7—figure supplement 2C*,**D**), the rate $f_{M,0}$ was set to zero. To mimic the deletion of the ASE motif from the *che-1* promoter (*Figure 7—figure supplement 2D*), we set the CHE-1 binding rate $f_{O,C}$ to zero.

## Estimation of required ON state lifetime

If spontaneous switches from the ON to the OFF state occur as a Poisson process with rate $r$ then the probability of a switching event occurring at time $t$ is given by the exponential distribution $p(t) = r \exp(-rt)$. The fraction of animals in which a switching event occurs before time $t = T$ is given by the cumulative distribution $P(t) = 1 - \exp(-rt)$. This fraction is smaller than a value $\phi$ if $r < -\ln(1-\phi)/T$. For small fractions $\phi \approx 0$, this can be approximated as $r < \phi/T$. If a switching event can occur only in 1 out of $10^6$ animals, $\phi = 1 \cdot 10^{-6}$, within the average animal lifetime of $T = 2$ weeks, then the required life time of the ON state is given by $1/r > 2.3 \cdot 10^5$ years.

## Predicting transcription factor binding

Candidate transcription factors were found using a combination of web-based tools: INSECT Tool 2.0 (*Parra et al., 2016*) and PROMO (using version 8.3 of TRANSFAC) (*Messeguer et al., 2002*). We used the 130 bp sequence which was deleted from the *che-1* promoter as input sequence. Here, apart from the ASE motif, the HD-binding site had the strongest predicted binding site. The binding sites were analysed more in depth with a custom written R script, using the TFBSTools and JASPAR 2018 packages, to recover the exact binding site and other nearby binding sites of candidate transcription factors.

## Auxin-induced protein degradation

The Auxin Inducible Degron (AID) System was employed as previously described (*Zhang et al., 2015*). Animals were initially grown on NGM plates with OP50 without IAA (Sigma-Aldrich). To induce degradation

of CHE-1::GFP::AID, L1 staged animals were transferred to NGM plates with OP50 and 1 mM IAA at 20 °C. IAA stocks were dissolved in 100% EtOH, leading to a final concentration of 0.2% EtOH in NGM plates. For the CHE-1::GFP::AID measurements, the duration of auxin exposure and recovery were varied in each treatment. The animals were transferred every other day to new NGM plates with 1 mM IAA or recovery NGM plates without IAA to prevent mixing of generations. Each treatment contained a control group of *che-1::GFP::AID* animals never exposed to auxin. For imaging, the animals were placed on an 5% agarose pad submerged in 0.25 mM Levamisole (Sigma-Aldrich) in M9 with a cover glass on top. We checked for CHE-1::GFP::AID in the ASE neurons with a wide field microscope. Older animals showed stronger autofluorescence in the head, giving rise to nuclei-like structures which could be confused with ASE neurons. The GFP (FITC) channel was combined with the TRITC channel to correct for the autofluorescence (*Teuscher and Ewald, 2018*). Due to tissue scattering and decrease in CHE-1::GFP::AID signal in old animals, we found the ASE neuron closest to the cover glass was most reliable. The control group always showed CHE-1::GFP::AID in the ASE neurons: even though CHE-1::GFP::AID levels decreased with age, even 10 days old animals had well-identifiable ASE neurons.

## NaCl chemotaxis assay

The quadrant assay used to assess chemotaxis to NaCl was adapted from *Jansen et al., 2002*; *Wicks et al., 2000*. Briefly, two diagonally opposite quadrants of a sectional petri dish (Star Dish, Phoenix Biomedical) were filled with 13.5 mL buffered agar (1.7% Agar, 5 mM $K_2HPO_4/KH_2PO_4$ pH 6, 1 mM $CaCl_2$ and 1 mM $MgSO_4$) containing 10 mM NaCl and two diagonally opposite quadrants with 13.5 mL buffered agar without NaCl. Immediately before the assay, the plastic dividers between the quadrants were covered with a thin layer of agar. Age synchronized young adult *C. elegans* populations were washed three times for 5 min with CTX buffer (5 mM $K_2HPO_4/KH_2PO_4$ pH 6, 1 mM $CaCl_2$ and 1 mM $MgSO_4$). Approximately 100 animals were placed in the middle of a sectional dish. After 10 min, animals on each quadrant were counted and a chemotaxis index (CI) was calculated for each plate (CI = (# animals on NaCl – # animals not no NaCl)/ total # animals). To determine the CI of a strain, two assays per day were performed on at least 2 different days.

To assess the effect of CHE-1::GFP::AID depletion on chemotaxis, animals were bleached and cultured for 24 hr on NGM plates without IAA. After 24 hr animals were transferred to NGM plates containing 1 mM IAA. To remove eggs and larvae, animals were washed using CTX buffer and a 30 μm pluriStrainer (pluriSelect) and transferred to a fresh NGM plate, with or without 1 mM IAA, starting at 96 hr into the experiment and repeated every 24 hr until the end of the experiment. After the experimental treatment duration, the chemotaxis index was determined. Subsequently, recovery was started by transferring animals to NGM plates without IAA and the chemotaxis index was determined 24 hr and 48 hr thereafter. At each timepoint, the chemotaxis index was determined in similarly aged *che-1(p679)* and untreated *che-1::GFP::AID* animals.

## Time lapse

Time-lapse imaging was performed as previously described (*Gritti et al., 2016*). Briefly, micro chambers are made out of polyacrylamide hydrogel, made from a 10% dilution of 29:1 acrylamide/bis-acrylamide (Sigma-Aldrich) was mixed with 0.1% ammonium persulfate (Sigma-Aldrich) and 0.01% TEMED (Sigma-Aldrich) as polymerization initiators. For the time-lapse experiments the chambers were 240 × 240 × 20 μm, these dimensions were sufficient to contain enough OP50 bacteria to sustain development until animals started laying eggs.

We used a Nikon Ti-E inverted microscope with a 40 X objective in all experiments. The microscope has a Hamamatsu Orca Flash 4.0 v2 camera set at full frame and full speed. The camera chip is 13 × 13 mm and contains 4Mp. We used 488 nm lasers (Coherent OBIS-LS 488–100) for fluorescence excitation. We used a high fluorescent signal of 100 mW with an exposure time of 10 ms, since the fluorescent signal of CHE-1::GFP is relatively low. To keep track of the molting cycle as indication of the age of the animals, we used bright field imaging, which contained a red LED (CoolLED, 630 nm). Time-lapse images were acquired every 20 min without detectable phototoxicity effects.

Images were analysed with custom written time-lapse software, and with ImageJ. Briefly, first we corrected the raw images for experimental aberrations with flat and dark field images acquired at the end of the experiment. For quantification purposes, we computed the average fluorescence of each

ASE neuron via drawing a region of interest around each nucleus and we corrected the average intensity by subtracting the background level close to the ASE neuron.

## Quantification and statistical analysis

Image analysis of CHE-1::GFP quantification and intensity measurements was performed with the ImageJ distribution Fiji (*Schindelin et al., 2012*). For the quantification data shown in graphs of all figures, the dots represent individual values, the boxplots without box represent the mean and the standard deviation.

Statistical analyses were performed either using R software, version 3.6.0, or with Python 3.5 Package SciPy. Comparisons of the chemotaxis indexes were performed using a one-way ANOVA, followed by a pairwise t-test with Holm correction. Significance between control versus conditions in smFISH data were preformed using one-way ANOVA, followed by a Tukey multiple comparison test. Significance of % expression of CHE-1::GFP in ASE neurons between treatment groups (on auxin) and control group (no auxin) was preformed using Fisher exact test.

## Acknowledgements

Some strains were provided by the CGC, which is funded by NIH Office of Research Infrastructure Programs (P40 OD010440), and the Mitani laboratory through the National Bio-Resource Project of the MEXT, Japan. We thank Dylan Rahe (Hobert lab) for providing the GFP-tagged *che-1(ot856)* allele. This work is part of the research program of the Foundation for Fundamental Research on Matter (FOM; grant FOM-161, 14NOISE05), which is financially supported by the Netherlands Organization for Scientific Research (NWO).

## Additional information

### Funding

| Funder | Grant reference number | Author |
| --- | --- | --- |
| Foundation for Fundamental Research on Matter | FOM Vrij Programma | Jeroen S van Zon |
| Foundation for Fundamental Research on Matter | FOM Vrij Programma | Gert Jansen |

The funders had no role in study design, data collection and interpretation, or the decision to submit the work for publication.

### Author contributions

Joleen JH Traets, Conceptualization, Data curation, Formal analysis, Investigation, Software, Writing – original draft, Writing – review and editing; Servaas N van der Burght, Conceptualization, Data curation, Formal analysis, Investigation, Resources, Validation, Writing – review and editing; Suzanne Rademakers, Investigation, Resources; Gert Jansen, Conceptualization, Funding acquisition, Supervision, Writing – review and editing; Jeroen S van Zon, Conceptualization, Funding acquisition, Methodology, Supervision, Writing – original draft, Writing – review and editing

### Author ORCIDs

Joleen JH Traets http://orcid.org/0000-0003-0505-9776
Servaas N van der Burght http://orcid.org/0000-0002-3272-3815
Gert Jansen http://orcid.org/0000-0002-7524-171X
Jeroen S van Zon http://orcid.org/0000-0002-6021-2924

### Decision letter and Author response

Decision letter https://doi.org/10.7554/eLife.66955.sa1
Author response https://doi.org/10.7554/eLife.66955.sa2

## Additional files

### Supplementary files

• Source code 1. Source code for Gillespie simulations underlying *Figure 4*, *Figure 4—figure supplement 1*, *Figure 7—figure supplement 2*.

• Supplementary file 1. Table showing recovery of CHE-1::GFP fluorescence and chemotaxis response on 10 mM NaCl in che-1::GFP::AID animals after auxin treatment.

• Supplementary file 2. CHE1 ASER/L: Genes containing CHE-1 binding sites (ASE motif) in genes expressed in the ASEL or ASER neuron. Genes expressed genes in ASER/L neurons were recovered based on single cell data from *Cao et al., 2017*. Each gene was tested on the presence of an ASE motif in the promoter (1000 bp upstream from the start site), with a binding site score of at least 98%. (OTX ASER/L) Genes containing homeodomain (HD) binding sites (OTX2 motif) in the same set of genes expressed in ASER/L neurons, with the same 98% cut-off for the binding site score. (OTX close to ASER/L) Genes that have an HD binding sites in close proximity (100 bp up- and/or down-stream) of any of the recovered CHE-1 binding sites.

• Supplementary file 3. Overview of single molecule FISH (smFISH) probes used for *che-1* and CHE-1 targets *gcy-22*, *tax-2* and *del-2*.

• Supplementary file 4. Overview of ssODNs and guides used in this research.

• Transparent reporting form

### Data availability

All analysed data and analysis scripts are included in the manuscript and supporting files.

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
