## [Editor Report]

This paper will be of interest to developmental biologists and neurobiologists who study the molecular mechanisms underlying induction and maintenance of cell fate. A combination of cutting-edge molecular genetic approaches in *C. elegans* together with mathematical modeling suggests an interesting mechanism for life-long maintenance of neuronal identity and function.

---

## [Decision Letter]

**Decision letter after peer review:**

Thank you for submitting your article "Mechanism of life-long maintenance of neuron identity despite molecular fluctuations" for consideration by *eLife*. Your article has been reviewed by 3 peer reviewers, and the evaluation has been overseen by a Reviewing Editor and Aleksandra Walczak as the Senior Editor. The following individual involved in review of your submission has agreed to reveal their identity: Attila Becskei (Reviewer #1).

Essential revisions:

(1) Additional experimental controls:

(1a) Age of animals needs to be controlled for

The different durations of treatment and recovery mean that animals are tested for ASE activity (using a chemotaxis assay) at different days of adulthood, ranging from two to six days after starting the experiment. Many cellular functions decline after a few days of adulthood (see for example Stein and Murphy, Frontiers in Genetics, 2012). Therefore, the author's conclusions could be influenced by a general decline in chemotaxis, and a general inability to recover CHE-1 expression in "older" adults.

It would be important to show a time course of NaCl chemotaxis through adulthood, covering the full range of ages tested in the different treatment regimes (the authors assume that CHE-1 stability and ASE function stay constant for the whole life of the worm, but this is unlikely to be the case). The authors may already have these data, but it is not very clearly presented.

In addition, if 3-4-day old adults are treated with auxin for 1-2 days, do they recover CHE-1 expression and ASE activity? If they cannot, this would rather suggest that the switch to the OFF state is a property of "old" ASEs. It is important to deconvolve the age component in these experiments.

(1b) Control for survival of ASE neurons required

In vertebrate systems (e.g. PMID: 30146154), inducible removal of a terminal selector in adult neurons often leads to cell death. It therefore seems critical to evaluate the percentage of ASE neurons that are still alive after auxin treatment and 48h recovery, in particular when there is no recovery. This control experiment will address whether the progressive decrease in the ability of CHE-1 to recover its protein levels (upon increased periods of auxin treatment) is related to ASE cell death.

(1c) Use of ethanol in controls required

Auxin is typically diluted in 0.25 % ethanol, and therefore 0.25% ethanol should be used in controls. Especially, since ethanol is known to affect animal physiology, gene expression and chemosensation. The current study seems to use "control" animals not exposed to 0.25% ethanol. The experiments shown in figures 1, 5, and 6 should include this important control. In line 133, the authors state that NaCl chemotaxis returned to wild-type levels after 24 hrs of auxin. However, based on Figure 1 and SFigure 1, NaCl chemotaxis did not quite return to wild-type levels, perhaps because the auxin-treated animals were exposed to ethanol, whereas the control animals were not.

(1d) Control for effect of the AID allele without auxin

The authors mutated the HD site in the context of the che-1::GFP::AID allele. Therefore, control che-1::GFP::AID animals with an intact binding site must be included in the analysis shown in Figure 7C, D, and E (and in Figure S5) to ensure that initiation of che-1 occurred normally in (ΔHD)p::che-1 animals. Previous studies have shown that the AID degron by itself (without addition of auxin) can generate hypomorphic effects that become severe over time (Kerk et al., 2017, PMID: 28056346). Hence, it is unclear whether the observed reduction in CHE-1::GFP::AID in (ΔHD)p::che-1 animals over time is an effect of mutating the HD site, or is caused by lowering the levels of CHE-1 due to the presence of the AID degron. If this control has been performed, it was not pointed out clearly enough.

(2) Substantiate the model or revise the conclusions:

Concerns about the current model were raised and need to be addressed. This could be through additional wet lab experiments, revisions to the model, changes in the text, or combinations of those.

Two weaknesses were pointed out in particular: (2a) that evidence for the CHE-1 reservoir is missing, and (2b) that the (direct) transcriptional autoregulation of che-1 underlying bistability is not (yet) well supported.

(2a) The evidence for the CHE-1 reservoir could, if technically possible, be strengthened by performing ChIP experiments to analyze whether CHE-1 still binds to its own promoter after induced CHE-1 depletion. Does CHE-1 relocate from the promoter of its target genes to its own promoter upon induced CHE-1 depletion? The authors state that crucial to this mechanism is that CHE-1 shows strong preferential binding to its own promoter compared to its other target genes, but this is somewhat contradictory to a previous study showing the affinity score of CHE-1 for its own promoters and its targets genes is similar (Etchberger et al., 2007). In the absence of any additional data, the conclusions should be toned down-for example in the abstract where the authors state "Fluctuations in CHE-1 level are buffered by the reservoir of CHE-bound at its target promoters".

(2b) It seems important to further clarify the mechanism of bistability:

The mathematical model describes a positive feedback through che-1 but does not take into account the highly relevant regulation by HD. A high feedback-independent (basal) expression of che-1 would preclude bistability even with marked nonlinearities ( Májer et al. 2015; Jaquet et al. 2017).

The che-1 mRNA remains fully expressed after 24 hours of auxin treatment despite the fact that che-1-GFP fluorescence disappears after 3 hours. Currently, the observations could also be explained by an alternative model in which the bistability arises in a feedback loop downstream of che-1, which would explain why the expression of target genes declines upon auxin treatment. This could be described as HD -> Che-1 -> Che-1 target genes and the latter ones generate bistability. Such a mechanism would be reminiscent of the GAL regulon in yeast (Acar et al., 2005). The Gal4 transcription factor activates the GAL target genes but the expression of Gal4 itself is not bistable. The bistability arises due to the regulators of Gal4 that feedback on the Gal4 activity.

This could be clarified by performing mRNA measurements also after a depletion lasting for 96 hours when the neuronal function (chemotaxis index) is fully lost.

If the che-1 mRNA level declines, the authors would need to update their model to separate the timescales of the che-1-dependent processes from the HD-dependent processes.

If the che-1 mRNA level does not decline even after 96 hours, there will be no evidence for a functional autoregulation of che-1 in the terminal state despite the presence of the che-1 binding site in the promoter. In this case, the mathematical model should be reduced to a minimum that would serve to explain the bistability and time series studies.

*Reviewer #1:*

Traets, van Zon and colleagues explore the determinants of the reversibility of neuronal cell fate determination due to the transcription factor che-1 in the worm *C. elegans*. For this purpose, they deplete the che-1 protein with an auxin-degron and follow the restoration of neuronal function after the discontinuation of the auxin treatment. The neuronal function, as measured by the chemotaxis index, is not restored provided the depletion period is long enough. At first glance, this experiment suggests that the autoregulation of che-1 is bistable.

Strengths:

1. The authors perform a transient depletion experiment, a quite useful method to detect bistability. The transient depletion experiment is a merit on its own since bistability is rarely detected (with appropriate methods) in the relevant literature. The transition to the off state upon the transient depletion indicates that che-1 is somehow involved in the bistability of ASE neuron cell fate determination.

2. The authors discover a new regulatory sequence the che-1 promoter targeted by the transcription factor HD, which is involved in cell fate determination.

3. They determined the values of che-1 parameters such as mRNA and protein half-lives. The methods they use are probably more reliable than the methods commonly used in the field.

Weakness:

1. The che-1 mRNA remains fully expressed even after 24 hours of auxin treatment despite the fact that che-1-GFP fluorescence disappears after 3 hours. This result suggests that there is bistability but it is not mediated through the (direct) transcriptional autoregulation of che-1 in the terminal neuronal state. However, the authors do model the direct positive feedback of the che-1 transcription factor. Bistability cannot arise in this system because of the high feedback-independent (basal) expression of che-1. The authors identify a new regulator of the che-1 promoter, the Otx-related transcription factor, which accounts for the high basal expression of the che-1 promoter in the terminal state.

All these observations could be explained by an alternative model in which the bistability arises in feedback loop downstream of the che-1, which would explain why the expression of target genes decline upon auxin treatment. Of course, it is possible that there is a bistable positive feedback through che-1 during the earlier stages of development but it becomes overshadowed by the Otx-related transcription factor (HD) in the terminal state, which is analyzed in the current experiments. Thus, the mathematical model and parts of the interpretation seem disconnected from the observations.

*Reviewer #2:*

In this manuscript the authors address the question of how a cell's identity is maintained even though it exists in a reversible, bi-stable state. Specifically, the ASE sensory neurons in *C. elegans* require sustained activity of the transcription factor CHE-1 throughout the life of the animal. CHE-1 autoregulates its own expression in a positive feedback loop, however, such positive-feedback loops can relatively easily switch between ON and OFF states. The authors show that indeed, both che-1 mRNA and CHE-1 protein have relatively short half-lives and are not present in great excess, meaning that normal fluctuations in gene expression could indeed result in spontaneous loss of CHE-1 expression. Using a CHE-1 degradation system to reduce CHE-1 for defined amounts of time, the authors show that CHE-1 expression can be subject to such bi-stability, raising the question of how this is prevented during the life of *C. elegans*. Using a combination of quantitative assays for mRNA and protein expression, precise genetic manipulations in vivo, as well as mathematical modeling, the authors propose a compelling explanation: even upon fluctuations that substantially reduce the level of CHE-1, the CHE-1 protein bound to its hundreds of targets provides a reservoir for continuous che-1 transcription, as CHE-1 binds preferentially to its own promoter relative to that of other targets.

The presented work is generally strong both conceptually and methodologically. The question is interesting and well-defined, the logic of the work is clear, and the methodology is of high quality. The main strength is the quantitative nature of the work, even more so considering this is all done in vivo. Overall, I think this paper provides insightful conclusions to generally interesting questions in gene regulation and cell identity. Below I raise one concern though, that requires an additional control in order to strengthen the conclusion that CHE-1 autoregulation is bi-stable.

The authors claim bi-stability of ASE identity and function by triggering degradation of CHE-1 for different lengths of time and asking whether CHE-1 and ASE activity recover, or CHE-1 switches to an OFF state. If CHE-1 is actively degraded with the auxin inducible system for one or two days, the ASE neurons lose identity and functionality; but after another day or two in the absence of auxin, both CHE-1 and ASE activity recover. However, if CHE-1 is degraded for three or four days, neither CHE-1 nor the ASE activity can be recovered, even after two days in the absence of auxin. The authors conclude that this shows that CHE-1 controls its own expression, and ASE identity, in a bi-stable manner.

There is one slight concern with these experiments and that is that according to my understanding, the different durations of treatment and recovery mean that animals are tested for ASE activity (using a chemotaxis assay) at different days of adulthood, ranging from two to six days after starting the experiment. Despite the authors often referring to *C. elegans* lifespan being about 2 weeks, many cellular functions decline after a few days of adulthood (see for example Stein and Murphy, Frontiers in Genetics, 2012). Therefore, the author's conclusions could be influenced by a general decline in chemotaxis, and a general inability to recover CHE-1 expression in "older" adults. The authors state that they assayed age-matched control animals, but these are not shown. A single control is shown in each panel, and it's unclear what the age of the control was.

In addition to a time-course of chemotaxis, it would be important to test whether older adults can recover CHE-1 expression (and ASE function) when faced with a shorter auxin treatment. Specifically, if 3-4-day old adults were treated with auxin for 1-2 days, could they recover CHE-1 expression and ASE activity? If they cannot, this would rather suggest that the switch to the OFF state is a property of "old" ASEs. This wouldn't invalidate the subsequent parts of the work, but would give a more accurate picture of what the contribution of the proposed mechanism is.

*Reviewer #3:*

This work studies how transcriptions factors control cell fate by focusing on terminal selectors – a type of transcription factors known to induce and maintain the identity of specific neuron types across species. Traditional studies have examined terminal selector function using mutant animals carrying alleles that eliminate gene activity from early development. Hence, it remains unclear how terminal selectors maintain neuronal identity in the adult animal in the context of post-mitotic neurons, which are inherently long-lived cells in all species. This paper combines cutting-edge molecular genetic approaches with mathematical modeling to study how the terminal selector CHE-1 maintains the identity of the chemosensory neuron ASE in the *C. elegans* nervous system. Previous studies have shown that CHE-1 is required to maintain its own expression, but the current paper examines whether such autoregulation is sufficient or additional mechanisms are involved for maintenance of ASE fate. To test this, the authors established an inducible system to deplete CHE-1 and assess effects on ASE function. They rigorously determined copy number and half-lives of che-1 mRNA and protein. Armed with this information, they performed sophisticated simulations of the CHE-1 switch to estimate its stability against stochastic fluctuations. These simulations led to the hypothesis that high stability of the ON state required that CHE-1 binds its own promoter stronger than that of its target genes ("target reservoir buffering" hypothesis), thereby making che-1 gene expression insensitive to stochastic decreases in CHE-1 protein level. Through precise genome engineering, the authors propose that an Otx-related homeodomain binding site is selectively responsible for che-1 maintenance, not initiation.

Additional analyses and controls are required to firmly test the hypothesis of "target reservoir buffering", which at present is not entirely supported by the experimental data.

Strengths:

This study employs cutting edge molecular, genetic and biophysical methods in combination with sophisticated modeling/simulations to study the molecular mechanism underlying maintenance of ASE fate.

The authors established a powerful system to deplete CHE-1 at will for different periods of time and then assess effects on ASE function, as well as on expression of che-1 itself and its target genes.

The authors go to great lengths (e.g., smFISH, FRAP) to determine copy number and half-lives of che-1 mRNA and protein.

They performed sophisticated simulations of the CHE-1 switch to estimate its stability against stochastic fluctuations. Such simulations gave rise to the interesting hypothesis of "target reservoir buffering".

One piece of data (Figure 5) strongly supports the hypothesis, albeit a single CHE-1 target gene was tested.

Elegant genome engineering identified a 130bp fragment responsible for che-1 maintenance when CHE-1 is depleted. Within this fragment, an Otx-related HD binding site is proposed to be responsible for che-1 maintenance.

Weaknesses:

The simulations do propose an interesting mechanism (target reservoir buffering), but this mechanism is only tested indirectly and for a single che-1 target gene (gcy-22). In addition, the conclusions would profit from additional controls: since in vertebrate systems inducible removal of a terminal selector in adult neurons often leads to cell death, it seems critical to evaluate the percentage of ASE neurons that are still alive after auxin treatment and recovery. In addition, control animals should be treated with the solvent for auxin, and effects of the AID-tag, independent of auxin treatment, should be tested.

[Editors' note: further revisions were suggested prior to acceptance, as described below.]

Thank you for re-submitting your article "Mechanism of life-long maintenance of neuron identity despite molecular fluctuations" for consideration by *eLife*.

Your article is provisionally accepted. Two of the reviewers, Paschalis Kratsios (Reviewer #3) and an anonymous reviewer, were fully satisfied by your revision, but we would like you to consider the additional comments by Attial Becskei (Reviewer #1). If you agree with his comments, this would mean slightly revising your discussion and Figure 7-S2. If you do not agree, please provide a justification.

Please see the detailed comments below. We look forward to receiving your re-submission very soon.

*Reviewer #1:*

The authors attempted to address the divergent behavior of CHE-1 mRNA and protein. The sampling period for the FISH experiments was extended. However, the background fluorescence was reported to increase over time and mRNAs could not be distinguished from the background. Next, they used RT-qPCR. However, they were not able to detect the CHE-1 mRNA with RT-qPCR. Thus, it remains unclear whether the CHE-1 mRNA level remains high level. These experiments would have been critical to distinguish two versions of the model.

At the same time, they rely heavily on the findings of Leyva-Diaz et al., Development, 2019 who report on the autoregulatory effects of CHE-1. In turn, they modify the model and off rate (unbinding rate) of CHE-1 is decreased 1000fold due to its interaction with HD-1. Unsurprisingly, such dramatic stabilization of the TF-DNA complex leads to a relative stabilization of the expression state. Consequently, the modelled "bistability" is stochastic, strongly time dependent. Future experiments will have to confirm this hypothesis.

Most bistability models in the literature rely on a deterministic bistability, which is then converted into stochastic model, whereas the stochastic component due to the slow dissociation rate is dominant in the author's model.

Bistability is prominently discussed in this manuscript. Therefore, the reader would gain a balanced view and profit from an extension of the discussion, in which deterministic bistability is compared to stochastic bimodality (bistability). For this, they can use the previously mentioned references and/or Hermsen et al. (2011) Plos Comp biol. Whereas kinetic nonlinearities and the dynamic range (basal expression) dominate deterministic bistability, the low number of molecules and time scales (e.g. off-rates) are key determinants of stochastic stability. The distinction also matters from a formal mathematical viewpoint. While quite general proofs can be derived for the existence of deterministic bistability, this is hardly ever the case for stochastic models. Generation of a few trajectories does not prove that a stochastic model is correct or incorrect. Therefore, I suggest replacing the labels "correct / incorrect" in Figure 7S2 by some more phenomenological terms, such as congruent / incongruent.

*Reviewer #2:*

The authors have done a careful and thorough revision. My previous questions and concerns are resolved and I fully support this manuscript for publication.

*Reviewer #3:*

The revised manuscript is very much improved. The authors have done a remarkable job addressing my comments by conducting new experiments and improving the text.

---

## [Author Response]

Essential revisions:(1) Additional experimental controls:(1a) Age of animals needs to be controlled forThe different durations of treatment and recovery mean that animals are tested for ASE activity (using a chemotaxis assay) at different days of adulthood, ranging from two to six days after starting the experiment. Many cellular functions decline after a few days of adulthood (see for example Stein and Murphy, Frontiers in Genetics, 2012). Therefore, the author's conclusions could be influenced by a general decline in chemotaxis, and a general inability to recover CHE-1 expression in "older" adults.It would be important to show a time course of NaCl chemotaxis through adulthood, covering the full range of ages tested in the different treatment regimes (the authors assume that CHE-1 stability and ASE function stay constant for the whole life of the worm, but this is unlikely to be the case). The authors may already have these data, but it is not very clearly presented.

We did already perform this control but did not show the data. Indeed, one would expect the chemotaxis to decline with age since many cellular functions decline over the course of the animal’s lifespan. However, we found that the chemotaxis index remained similar over the range of ages (48-168 hrs. after hatching) we probed experimentally.

Changes to the manuscript:

– We now show this new result in Figure S2D and added the text “The long duration … off auxin (Figure S2E).” to discuss this result in the section “Loss of ASE neuron fate upon transient CHE-1 depletion“ in the Results.

– Because of the addition of a number of new experiments, we split the old Figure S1 into two new supplementary figures, with one focusing on CHE-1::GFP::AID (Figure S1) and one on chemotaxis assays (Figure S2).

In addition, if 3-4-day old adults are treated with auxin for 1-2 days, do they recover CHE-1 expression and ASE activity? If they cannot, this would rather suggest that the switch to the OFF state is a property of "old" ASEs. It is important to deconvolve the age component in these experiments.

We confirmed that 72 hrs old adults that were subsequently treated with auxin for 48 hrs do recover NaCl chemotaxis 24 and 48 hrs after the end of induced CHE-1 depletion. Hence, the observed failure of animals to recover from transient CHE-1 depletion did not reflect an increasing bias towards the OFF state due to aging.

Changes to the manuscript:

We have included this data in Figure S2E and added text “The long duration … off auxin (Figure S2E).” to discuss this result in the section “Loss of ASE neuron fate upon transient CHE-1 depletion“ in the Results.

(1b) Control for survival of ASE neurons requiredIn vertebrate systems (e.g. PMID: 30146154), inducible removal of a terminal selector in adult neurons often leads to cell death. It therefore seems critical to evaluate the percentage of ASE neurons that are still alive after auxin treatment and 48h recovery, in particular when there is no recovery. This control experiment will address whether the progressive decrease in the ability of CHE-1 to recover its protein levels (upon increased periods of auxin treatment) is related to ASE cell death.

We agree that this is an important control, that we originally did not perform. A challenge is that most ASE-specific markers are transcriptional reporters of CHE-1 target genes and might therefore cease expression upon sufficiently long CHE-1 depletion. We now ruled out that ASE cell death is responsible for the lack of CHE-1 recovery upon CHE-1 depletion, in the following manner. We crossed an *osm-3::GFP* reporter into *che-1::GFP::AID* animals. OSM-3 functions in ciliated neurons and is expressed in 10 pairs of amphid neurons, including the ASE neurons. We therefore expected that *osm-3* expression would not be lost upon CHE-1 depletion. When we exposed these animals to auxin for 108 hrs (an exposure for which almost all animals fail to recover CHE-1, see Figure 1F) and counted neurons on one side of the animal, we indeed found 10 cells in n=26 animals. This shows that ASE neurons did not die even after prolonged CHE-1 depletion.

Changes to the manuscript:

– We added panels Figure S1D as illustration of the *osm-3::GFP* experiments.

– We added the text “In mice, the removal.… after prolonged CHE-1::GFP::AID depletion” to the section ‘Loss of ASE neuron fate upon transient CHE-1::GFP::AID depletion’.

– To describe the generation of the *osm-3::GFP* strain in the Methods section, we added a new section “Molecular biology” and we added the text “To generate the *osm-3::GFP*…and screened by PCR” to the section “CRISPR/Cas9-mediated genome editing”.

(1c) Use of ethanol in controls requiredAuxin is typically diluted in 0.25 % ethanol, and therefore 0.25% ethanol should be used in controls. Especially, since ethanol is known to affect animal physiology, gene expression and chemosensation. The current study seems to use "control" animals not exposed to 0.25% ethanol. The experiments shown in figures 1, 5, and 6 should include this important control. In line 133, the authors state that NaCl chemotaxis returned to wild-type levels after 24 hrs of auxin. However, based on Figure 1 and SFigure 1, NaCl chemotaxis did not quite return to wild-type levels, perhaps because the auxin-treated animals were exposed to ethanol, whereas the control animals were not.

We used stock solutions with Auxin dissolved in 100% ethanol, leading to a concentration of ~0.25% ethanol in our NGM+auxin plates. Our experimental controls in Figures 1, 5 and 6 were indeed performed with wild-type N2 animals on NGM plates without ethanol (Figures 1, 6) or with *che-1::GFP::AID* animals on NGM plates without auxin and ethanol (Figures 1, 5). The experiments in Figure 1 concern mostly chemotaxis assays, those in Figure 5 gene expression measurements and Figure 6 has both chemotaxis and gene expression measurements. We agree with the reviewers that it is important to rule out an impact of ethanol by itself on chemotaxis or gene expression.

For the chemotaxis assays, we performed new controls on wild-type animals with 0.25% ethanol and found no impact on NaCl chemotaxis of 24, 72 and 96 hours culture on ethanol (Figure S2B). For gene expression, we made new measurements of the *che-1* mRNA copy number in animals on 0.25% ethanol. We found that *che-1* mRNA levels were similar to those found in wild-type animals and in *che-1::GFP::AID* animals without auxin and 0.25% ethanol (Figure S3C).

Changes to the manuscript:

– We added the new 0.25% ethanol chemotaxis control experiment to Figure S2B, which we refer to in the section “Loss of ASE neuron fate upon transient CHE-1 depletion” when we conclude that the decrease in NaCl chemotaxis upon exposure to auxin is significant.

– We added the new gene expression measurements with 0.25% ethanol to Figure S3C, and added the text “As auxin is dissolved … from CHE-1::GFP::AID depletion” to the section “in vivo CHE-1 depletion decreases target gene but not *che-1* expression”.

(1d) Control for effect of the AID allele without auxinThe authors mutated the HD site in the context of the che-1::GFP::AID allele. Therefore, control che-1::GFP::AID animals with an intact binding site must be included in the analysis shown in Figure 7C, D, and E (and in Figure S5) to ensure that initiation of che-1 occurred normally in (ΔHD)p::che-1 animals. Previous studies have shown that the AID degron by itself (without addition of auxin) can generate hypomorphic effects that become severe over time (Kerk et al., 2017, PMID: 28056346). Hence, it is unclear whether the observed reduction in CHE-1::GFP::AID in (ΔHD)p::che-1 animals over time is an effect of mutating the HD site, or is caused by lowering the levels of CHE-1 due to the presence of the AID degron. If this control has been performed, it was not pointed out clearly enough.

It is correct that the HD site was removed in the *che-1::GFP::AID* background. We agree with the reviewers that to show that the lower CHE-1::GFP level and the spontaneous loss of ASE fate are due to the ΔHD mutation, we have to rule out that they are instead caused by the presence of AID degron itself.

A number of lines of evidence, based on our existing data and new control experiments, show that the AID tag itself, without auxin, has no adverse effect on maintenance of ASE fate and *che-1* expression levels:

1) Figure 7C shows that over time the percentage of *(ΔHD)p::che-1::GFP::AID* animals that exhibit CHE-1::GFP::AID expression decreases. As a control, we have now scored the fraction of animals that express CHE-1::GFP::AID in ASE neurons for *che-1::GFP::AID* young adults without auxin. We found no loss of CHE-1::GFP::AID in any ASE neuron in 23 animals followed until the young adult stage, in contrast to the data for *(ΔHD)p::che-1::GFP::AID* animals in Figure 7C. This means that spontaneous loss of ASE fate only occurred in *(ΔHD)p::che-1::GFP::AID* animals and not in *che-1::GFP::AID* animals.

2) Figure 7D shows reduced *che-1* mRNA levels in *(ΔHD)p::che-1::GFP::AID* animals. The *che-1::GFP:AID* control for this experiment is found in Figure 5B. Figure 5B shows that *che-1* mRNA copy number in *che-1::GFP:AID* is higher than in *(ΔHD)p::che1::GFP::AID* animals and of the same level as in wild-type animals (Figure 2B). This shows that the AID tag itself is not responsible for lower *che-1* mRNA levels.

3) Figure 7E shows reduced CHE-1::GFP::AID protein levels in *(ΔHD)p::che-1::GFP::AID* animals. We have now performed new measurements of the absolute copy number of CHE-1::GFP::AID protein (Figure S6B) and found that it was similar to that of CHE1::GFP and higher than the copy number observed in *(ΔHD)p::che-1::GFP::AID* animals.

4) We also performed new time-lapse measurements in *che-1::GFP::AID* animals, as control for the CHE-1::GFP time-lapse data in Figure 7E, showing a similar fluorescence level for both strains (Figure S6E).

Overall, these experiments show that the AID tag does not lead to lower CHE-1 mRNA and protein levels.

Changes to the manuscript:

We made the following edits to the section “Involvement of an Otx-related homeodomain binding site in maintaining ASE subtype”:

– For (1), we have added the text “while CHE-1::GFP::AID was … young adults (n=23 animals)”, which gives the result of our new control experiments

– For (2-4), we added the text “both considerably lower … animals (Figure 2B, 5B, S6B,E)” to point both to the control results presented earlier in the manuscript and to the new experiments we performed.

– For (3) and (4), we updated Figure S6B to include our new CHE-1::GFP copy number measurements in *che-1::GFP::AID* animals, as well as our original measurements in wildtype animals. We added Figure S6E to show time-lapse measurements in *che-1::GFP::AID* animals.

(2) Substantiate the model or revise the conclusions:Concerns about the current model were raised and need to be addressed. This could be through additional wet lab experiments, revisions to the model, changes in the text, or combinations of those.Two weaknesses were pointed out in particular: (2a) that evidence for the CHE-1 reservoir is missing, and (2b) that the (direct) transcriptional autoregulation of che-1 underlying bistability is not (yet) well supported.(2a) The evidence for the CHE-1 reservoir could, if technically possible, be strengthened by performing ChIP experiments to analyze whether CHE-1 still binds to its own promoter after induced CHE-1 depletion. Does CHE-1 relocate from the promoter of its target genes to its own promoter upon induced CHE-1 depletion?

The suggested ChIP-seq experiment would indeed be highly valuable to explicitly test these predictions from the target reservoir buffering. However, despite its importance as a canonical cell fate regulator, there are no existing CHE-1 ChIP-seq datasets we could currently build on. This likely reflects that it is technically very challenging to do ChIP experiments on CHE-1, due to the low levels of CHE-1 protein (~2000 copies in only two ASE neurons within the body of an animal). Even though this is something we hope to pursue in the future, for exactly the reasons outlined by the reviewers, we believe that overcoming these challenges falls outside of the scope of this work.

Changes to the manuscript:

– In the Discussion, we discuss these issues in the section “A definitive test … neurons

(Kaletsky et al., 2016).”

The authors state that crucial to this mechanism is that CHE-1 shows strong preferential binding to its own promoter compared to its other target genes, but this is somewhat contradictory to a previous study showing the affinity score of CHE-1 for its own promoters and its targets genes is similar (Etchberger et al., 2007).

Regarding preferential binding of CHE-1 to its own promoter: we speculated in the paper that the HD motif recruits a co-factor that leads to stronger binding of CHE-1 by cooperative interactions with the co-factor. We now added simulations (see our response to (2b) below) that demonstrate the validity of this idea. The affinity scores of the ASE site would not take into account this cooperative interaction. Indeed, our observations that the HD binding site is only found in the *che-1* promoter and that swapping ASE sites between *che-1* and *gcy-22* has no impact on resilience of *che-1* expression to CHE-1 depletion are all consistent with the similar affinity scores found in Etchberger et al.

In the absence of any additional data, the conclusions should be toned down-for example in the abstract where the authors state "Fluctuations in CHE-1 level are buffered by the reservoir of CHE-bound at its target promoters".

We identified the target reservoir buffering mechanism in the context of our simulations. While we were able to experimentally confirm a key prediction of the mechanism (the resilience of *che-1* expression to CHE-1 depletion, while other *che-1* targets display no such resilience, as shown in Figures 4E, 5), we agree with the reviewer that we cannot provide experimental evidence on the level of CHE-1 binding affinity. We have rewritten the manuscript, including the abstract, to clarify the distinction between the conclusions we draw from our simulations and those we draw from our experimental data.

Changes to the manuscript:

– In the abstract, we have rewritten the section “Our simulations identified … experimental evidence for this mechanism…” to clarify that the target reservoir mechanism was identified in the context of simulations and that a number of key predictions following from the model were subsequently confirmed experimentally.

– In the last paragraph of the Introduction, we explicitly distinguish between conclusions from simulations and experiments: “Our simulations revealed…” and “Consistent with this mechanism, we observed…”

– In the second paragraph of the Discussion, we similarly write: “Instead, our simulations suggested a novel mechanism…” and “Our experimental observations verified a key prediction…”

– In the third paragraph of the Discussion, we discuss the limits of our experimental evidence: “A definitive test of the target reservoir buffering mechanism would require…”

(2b) It seems important to further clarify the mechanism of bistability:The mathematical model describes a positive feedback through che-1 but does not take into account the highly relevant regulation by HD. A high feedback-independent (basal) expression of che-1 would preclude bistability even with marked nonlinearities ( Májer et al. 2015; Jaquet et al. 2017).

We agree with the reviewers that a high basal rate of *che-1* expression (here suggested to occur due to the presence of HD-TF) would ultimately break bistability. However, we propose an alternative model, where HD-TF increases the affinity of CHE-1 for the *che-1* promoter by cooperative interactions, but does not induce *che-1* expression itself. See the point directly below for more discussion.

The che-1 mRNA remains fully expressed after 24 hours of auxin treatment despite the fact that che-1-GFP fluorescence disappears after 3 hours. Currently, the observations could also be explained by an alternative model in which the bistability arises in a feedback loop downstream of che-1, which would explain why the expression of target genes declines upon auxin treatment. This could be described as HD -> Che-1 -> Che-1 target genes and the latter ones generate bistability. Such a mechanism would be reminiscent of the GAL regulon in yeast (Acar et al., 2005). The Gal4 transcription factor activates the GAL target genes but the expression of Gal4 itself is not bistable. The bistability arises due to the regulators of Gal4 that feedback on the Gal4 activity.

A model where *che-1* expression is maintained not by CHE-1, but instead by one of its target genes, would go against a large body of work on ASE fate specification (in particular, Etchberger 2007 and 2009 and Leyva-Diaz 2019). However, we agree with the reviewers that in our original manuscript we never tested our proposed mechanism of action of the HD-TF, i.e. that it increases the residence time of CHE-1 on the *che-1* promoter by acting as a co-factor, explicitly by simulations. Therefore, we have performed new simulations that explicitly take into account the regulation by the HD-TF (Figure S7) and also allowed us to test the alternative model proposed by the reviewers above.

We examined two possible mechanisms of HD-TF action: (1) HD-TF increases CHE-1 binding by cooperative interactions (our current preferred model, see our discussion in (2a) above) and (2) HD-TF can induce expression of *che-1* independently of CHE-1, which corresponds to the mode of action proposed here by the reviewer. We also allowed for the possibility that in both models the HD-TF would be a target of CHE-1 itself. With this assumption, for Model 2 the CHE-1 network would map more or less directly onto the GAL regulon network mentioned by the reviewer.

However, our simulations show that model (2), while it would explain the resilience of *che-1* expression as the reviewer suggests, is not consistent with other experimental data in the CHE-1 literature. Specifically, it predicts that once CHE-1 is induced during development, binding of CHE-1 to its own promoter is no longer needed for *che-1* expression. This is at odds with the experimental observation that if the ASE site in the *che-1* promoter is deleted, CHE-1 is expressed at high level during induction in embryos, but then vanishes once the inductive signal is downregulated at the end of embryogenesis (Leyva-Diaz et al., Development, 2019). Instead, model (2) predicts that upon removal of the ASE binding site, *che-1* expression remains after the inductive signal has disappeared (Figure S7D). We believe a similar line of reasoning would invalidate other models in which bistability in *che-1* expression is not due to positive feedback of CHE-1 but instead of one or more of its targets.

Changes to the manuscript:

– We added a new figure (Figure S7) to present our novel simulations that explicitly incorporate the interactions of HD-TF with CHE-1 and the *che-1* promoter.

– We added the text “In contrast, an alternative model.… specifically for its own promoter” to the section “Involvement of an Otx-related homeodomain binding site in maintaining ASE subtype” to discuss the results of our new simulations.

– We edited the text “Our theoretical estimates of ON state lifetimes.… interaction of a few kBT” in the Discussion to incorporate the new simulation results.

– We added an overview of the HD-TF model and the parameters of these new simulations to the Methods.

This could be clarified by performing mRNA measurements also after a depletion lasting for 96 hours when the neuronal function (chemotaxis index) is fully lost.If the che-1 mRNA level declines, the authors would need to update their model to separate the timescales of the che-1-dependent processes from the HD-dependent processes.

We believe we now address this issue regarding the model with our new simulations in Figure S7. Here, we directly model HD-TF and CHE-1 binding to the *che-1* promoter separately and the higher affinity of CHE-1 for its own promoter now arises explicitly from cooperative interactions between CHE-1 and HD-TF (Figure S7A, model 1)

If the che-1 mRNA level does not decline even after 96 hours, there will be no evidence for a functional autoregulation of che-1 in the terminal state despite the presence of the che-1 binding site in the promoter. In this case, the mathematical model should be reduced to a minimum that would serve to explain the bistability and time series studies.

We agree with the reviewer that this would be a great experiment: our model would predict that during the CHE-1 depletion, *che-1* mRNA would remain at its normal level until, driven by fluctuations, it rapidly falls to zero. At that point, CHE-1 will fail to recover when CHE-1 depletion is removed.

However, the main reason that we did not perform this experiment is the challenge of quantifying *che-1* mRNA levels in animals at 96 hours: we find both that the smFISH signal weakens as animals develop into adulthood (likely due to light scattering in these larger bodies) and that the autofluorescence background increases as animals age. For these combined reasons, we have never been able to detect *che-1* mRNA by smFISH in 96 hr animals. We tried to overcome this limitation in two ways: we performed *che-1* RT-qPCR on animals on auxin for 96 hrs and we stained *che-1* by smFISH in *che-1::GFP::AID* animals put on auxin for 40 hrs directly after hatching.

Unfortunately, we were not able to detect *che-1* expression by RT-qPCR in animals at any stage. This is likely due to its low copy number, as we readily detected transcripts of control genes.

For animals at 48 hrs, a substantial fraction of individuals already failed to recover CHE-1 (Figure 1F). We therefore expected that a fraction of these animals would show no *che-1* mRNA by smFISH, while the rest would show wild-type levels. While we did observe animals without *che-1* mRNA after 48 hrs on auxin (4/22 animals), we also found animals where we could not detect *che-1* mRNA in 48 hrs animals without auxin (1/15 animals). This is something we almost never see in animals off auxin that are younger than 48 hrs, and is consistent with the decreased quality of smFISH staining for these older animals: we believe that we are missing weak smFISH spots that would be visible under better imaging conditions.

So, overall, while the additional experiments we performed certainly do not rule out the mechanism of bistability that we propose (and that is the current working model in the field), these experiments unfortunately are not able to provide additional experimental evidence. However, supported by our new simulations in Figure S7, we are confident that a model where bistability in *che-1* level is controlled by feedback loops independent of CHE-1, as proposed by the reviewer, is unlikely. For that reason, we strongly believe that presenting the current model, which incorporates the key CHE-1 interactions as reported in the literature in a relatively detailed manner, is justified, and more useful to the community than a more abstract minimal mode, as suggested by the reviewer.

[Editors' note: further revisions were suggested prior to acceptance, as described below.]

Reviewer #1:The authors attempted to address the divergent behavior of CHE-1 mRNA and protein. The sampling period for the FISH experiments was extended. However, the background fluorescence was reported to increase over time and mRNAs could not be distinguished from the background. Next, they used RT-qPCR. However, they were not able to detect the CHE-1 mRNA with RT-qPCR. Thus, it remains unclear whether the CHE-1 mRNA level remains high level. These experiments would have been critical to distinguish two versions of the model.At the same time, they rely heavily on the findings of Leyva-Diaz et al., Development, 2019 who report on the autoregulatory effects of CHE-1. In turn, they modify the model and off rate (unbinding rate) of CHE-1 is decreased 1000fold due to its interaction with HD-1. Unsurprisingly, such dramatic stabilization of the TF-DNA complex leads to a relative stabilization of the expression state. Consequently, the modelled "bistability" is stochastic, strongly time dependent. Future experiments will have to confirm this hypothesis.Most bistability models in the literature rely on a deterministic bistability, which is then converted into stochastic model, whereas the stochastic component due to the slow dissociation rate is dominant in the author's model.Bistability is prominently discussed in this manuscript. Therefore, the reader would gain a balanced view and profit from an extension of the discussion, in which deterministic bistability is compared to stochastic bimodality (bistability). For this, they can use the previously mentioned references and/or Hermsen et al. (2011) Plos Comp biol. Whereas kinetic nonlinearities and the dynamic range (basal expression) dominate deterministic bistability, the low number of molecules and time scales (e.g. off-rates) are key determinants of stochastic stability. The distinction also matters from a formal mathematical viewpoint. While quite general proofs can be derived for the existence of deterministic bistability, this is hardly ever the case for stochastic models. Generation of a few trajectories does not prove that a stochastic model is correct or incorrect. Therefore, I suggest replacing the labels "correct / incorrect" in Figure 7S2 by some more phenomenological terms, such as congruent / incongruent.

We believe that Reviewer #1’s main objection is based on a misunderstanding of the extended model that incorporates the homeodomain transcription factor (HD-TF).

The Reviewer worries that our model exhibits stochastic bimodality rather than ‘deterministic’ bistability. Deterministic bistability means that the system can exist in two stable states when considering the macroscopic rate equations. Stochastic bimodality (as discussed in Hermsen et al. 2011) means that the stochastic system flips between two states where the dynamics lingers, even though these states are not stable states when considering the macroscopic rate equations. It is therefore a purely stochastic phenomenon and does not reflect ‘intrinsic’ bistability in the network. In this case, showing individual trajectories, as we do e.g. in Figure 7S-2C, would indeed be misleading, as many other simulations might fail to maintain CHE-1 expression upon transient CHE-1 induction.

However, this is explicitly not the case for our model. Author response image 1 shows transient induction (comparable to Figure 7S-2C) for the macroscopic rate equations of the original CHE-1 model (left) and the extended model with HD-TF (right). In both cases, transient CHE-1 induction (grey area) results in a stable switch from the low to the high CHE-1 state, thereby demonstrating the HD-TF model shows ‘true’ bistability, not just stochastic bimodality. In Figure 7S-2, we wanted to show the stochastic simulations, not the solutions to the rate equations shown below, to highlight that the added stochasticity (e.g. of HD-TF binding) did not negatively impact the stability of the high CHE-1 state. Overall, these stochastic simulations reproduce the solution to the rate equations very well. We realized that we did not explicitly make the point in our manuscript that the HD-TF models are deterministically bistable and have now added this observation to the text.

**Author response image 1. sa2fig1:** 

The Reviewer raises the possibility that the 1000-fold lower CHE-1 unbinding rate (when in a complex with HD-TF) stochastically ‘traps’ the system in a long-lived state, resulting in stochastic bimodal behavior even in absence of deterministic bistability. However, the 1000-fold lower unbinding rate still corresponds to an average lifetime of the CHE-1-HD-TF complex of only 10 seconds, meaning that the lifetime of the high CHE-1 state seen in the stochastic simulations in Figure 7S-2C is not the result of a single CHE-1-HD-TF remaining bound on the che-1 promoter for >10 hours. Hence, the Reviewer’s suggestion that the dynamics of our model is dominated by the slow CHE-1 dissociation rate appears not correct.Changes to the manuscript:

– We have modified the text “In this model, we.… reproduce three key experiments” in the section “Involvement of an Otx-related homeodomain binding site in maintaining ASE fate”, to point out that on the level of mass action rate equations the HD-TF model is bistable. Because all HD-TF models we consider are deterministically bistable, we did not add a discussion on bistability versus stochastic bimodality, as requested by the Reviewer.

– Similarly, we added the sentence “All four models exhibit bistability in their mass action rate equations” to the caption of Figure 7S-2.

– In Figure 7S-2, we changed “(in)correct” into the more phenomenological “(in)consistent”.